# Elemental mapping in single-particle reconstructions by reconstructed electron energy-loss analysis

Olivia Pfeil-Gardiner ●[1], Higor Vinícius Dias Rosa ●[1,5], Dietmar Riedel ●[2], Yu Seby Chen ●[3], Dominique Lörks[4], Pirmin Kükelhan[4], Martin Linck[4], Heiko Müller[4], Filip Van Petegem[3] & Bonnie J. Murphy ●[1] ✉

For macromolecular structures determined by cryogenic electron microscopy, no technique currently exists for mapping elements to defined locations, leading to errors in the assignment of metals and other ions, cofactors, substrates, inhibitors and lipids that play essential roles in activity and regulation. Elemental mapping in the electron microscope is well established for dose-tolerant samples but is challenging for biological samples, especially in a cryo-preserved state. Here we combine electron energy-loss spectroscopy with single-particle image processing to allow elemental mapping in cryo-preserved macromolecular complexes. Proof-of-principle data show that our method, reconstructed electron energy-loss (REEL) analysis, allows a three-dimensional reconstruction of electron energy-loss spectroscopy data, such that a high total electron dose is accumulated across many copies of a complex. Working with two test samples, we demonstrate that we can reliably localize abundant elements. We discuss the current limitations of the method and potential future developments.

In cryogenic electron microscopy (cryo-EM) single-particle analysis (SPA), experimental densities are interpreted to atomic models of macromolecular complexes with the help of substantial prior information. This comprises knowledge of peptide or nucleotide sequence, bond lengths and angles, and secondary structure. For protein and nucleotide components, individual atoms can thus be accurately placed, even though experimental data seldom reach atomic resolution and these atoms can be identified, in spite of the fact that no tools exist for elemental mapping. For all other components of a structure, including bound cofactors, lipids, substrates and inhibitors, post-translational modifications and, especially, metals and other ions, prior information is less easily available. For these, the interpretation of experimental data is thus often ambiguous, leading to errors in atomic models.

A recent study, analyzing 30 randomly selected metalloproteins, whose X-ray crystal structures had been deposited in the Protein Data Bank (PDB), found that one-third to one-half of the structures contain errors in metal assignment[1]. For structures determined by SPA cryo-EM, which in 2023 encompassed a third of all deposited structures[2], the error rate is probably higher, since there is currently no method that retrieves elemental information in the spatial context of the structure—a challenge we aim to address.

Methods for bulk elemental analysis of purified complexes (X-ray fluorescence or absorbance, optical emission spectroscopy and inductively coupled plasma mass spectrometry) cannot distinguish between specifically bound and soluble components and are inadequate for complexes with multiple binding sites. For structures

[1]Redox and Metalloprotein Research Group, Max Planck Institute of Biophysics, Frankfurt, Germany. [2]Facility for Electron Microscopy, Max Planck Institute for Multidisciplinary Sciences, Göttingen, Germany. [3]The University of British Columbia, Vancouver, British Columbia, Canada. [4]CEOS GmbH, Heidelberg, Germany. [5]Present address: Mattei Lab, Structural and Computational Biology Unit, EMBL Heidelberg, Heidelberg, Germany. ✉e-mail: bonnie.murphy@biophys.mpg.de

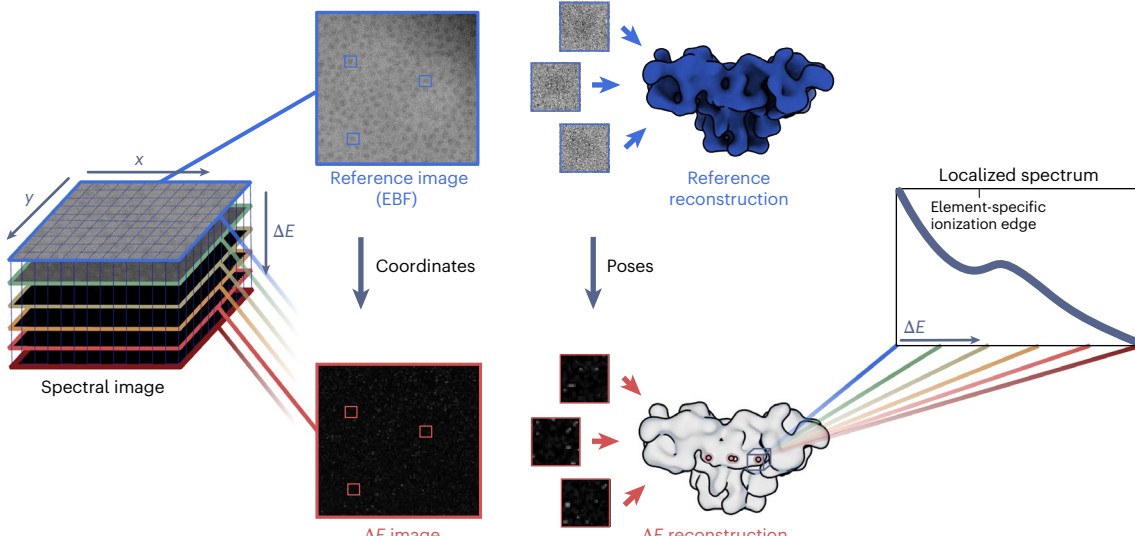

**Fig. 1 | Schematic of the REEL analysis workflow.** Spectral images of cryo-preserved macromolecular complexes are collected in the STEM-EELS mode. In addition to two spatial dimensions ($x$ and $y$), these contain an energy-loss dimension ($\Delta E$). From these images, the zero-loss portion is extracted to form elastic bright-field (EBF) images, serving as reference images, from which particle positions and reconstruction poses can be determined by refining a reference reconstruction. Then, $\Delta E$ images are extracted for all energy bins along the energy-loss spectrum. The particle coordinates and reconstruction poses determined from the reference are then applied to each of these images—indicated here at one exemplary energy loss—such that a volume can be reconstructed for every energy loss, generating a 4D data set with three spatial dimensions and one energy-loss dimension. In addition to analysis of the reconstructions, localized spectra can be generated by plotting the values of specific voxels or voxel regions as a function of energy loss.

determined by X-ray crystallography, the anomalous scattering signal at specific wavelengths offers a solution for many elements, though not for light elements (such as C, N, O, Na, Mg and P) that are most abundant in biological samples. However, not all samples are amenable to crystallization.

In the electron microscope, elemental information is available from inelastic scattering processes by means of energy-dispersive X-ray spectroscopy (EDX) and electron energy-loss spectroscopy (EELS)—both of which are recorded in scanning mode—or the corresponding imaging mode, electron spectroscopic imaging (ESI). These techniques have rarely been used for radiation-sensitive biological samples, as very high doses are required to accumulate sufficient signal from inelastic scattering events. Typically, high-resolution studies of biological samples are performed at total electron doses (more accurately, fluences) of 30–60 e⁻ Å⁻² to limit the degradation of structural features due to radiation damage. A study evaluating the sensitivity of EELS for biological samples estimated that $10^7$ e⁻ Å⁻² would be required for single-atom detection[3]. Such a dose would leave little-to-no structural context intact to be analyzed.

In cryo-preserved cell and tissue samples, abundant elements have been mapped by EDX, EELS and ESI[4–12]. In one instance, the assignment of an unknown cofactor in a cryo-preserved protein sample was achieved on the basis of bulk EELS without any spatial information[13]. Only a few studies have evaluated three-dimensional (3D) elemental mapping by electron microscopy in macromolecular complexes[14–17].

In SPA, two-dimensional (2D) images with low signal-to-noise ratio (SNR) are combined into a single 3D reconstruction with a high SNR. A similar reconstruction technique could conceivably be applied to accumulate spatially resolved elemental signals. Early studies of freeze-dried ribosomes and nucleosomes applied SPA to ESI images at the phosphorus edge to map nucleic acids in these complexes[14,15]. However, the high doses necessary to estimate particle poses from ESI images are incompatible with cryo-preserved samples or high resolution. Another study performed on inorganic nanoparticles used single-particle reconstruction of EDX data for mapping of elemental distributions but was likewise performed at high doses under noncryogenic conditions[18]. A separate study demonstrated the localization of metal ions within a protein reconstruction exploiting the differences of light and heavy elements in high-angle scattering intensities but did not discriminate between different heavy elements in a single reconstruction[16].

We propose a method that we call reconstructed electron energy-loss (REEL) analysis for a holistic elemental description of a macromolecular complex in three dimensions. In this approach, we combine scanning transmission electron microscopy (STEM)-EELS with an SPA-like workflow, in which signal from spectral images of many copies of a macromolecule is accumulated by a 3D reconstruction to meet the high-dose requirements for elemental mapping. A key difference to previous studies is that SPA particle poses are estimated on the basis of reference images formed by elastic scattering rather than from the energy-loss images themselves. Using these poses, 3D reconstructions can be generated for the full energy-loss spectrum, allowing analysis of elemental signatures in a 3D spatial context (Fig. 1) at doses compatible with cryo-preserved biological samples. To usefully address many biological questions, the technique would need to reach single-atom sensitivity. Although high resolution would also be desirable, even at low resolution, the elemental signal can be overlaid on higher-resolution densities from standard SPA to inform the process of atomic model building. Therefore, the required resolution depends strongly on the individual biological question to be addressed.

Here, we establish a workflow for data collection and processing for REEL analysis. Using this workflow, we provide proof-of-principle data demonstrating the mapping of specific elemental signals to the 3D space of a macromolecular reconstruction. We investigate the potential and current limitations of the method.

## Results

To enable elemental mapping of dose-sensitive complexes by REEL analysis, we have established the following workflow (Fig. 1). We collect spectral images of a conventionally prepared single-particle sample, operating the microscope in STEM-EELS mode at doses below 100 e⁻ Å⁻². We sum intensities of the zero-loss portion of the spectral images to form high-intensity reference images that we use to determine particle coordinates and reconstruction poses. These coordinates and poses

are subsequently used to reconstruct 3D volumes at each energy bin across the full energy-loss spectrum. The result is a four-dimensional (4D) data set containing three spatial and one energy-loss dimension(s). This can be viewed as a series of reconstructions or as one reconstruction with a spectrum at each voxel. By analyzing reconstructions at different energies or spectra at different locations, elements can be identified and mapped to spatial regions of a protein.

## A hardware configuration for cryo-STEM-EELS data collection

We carried out this work on a high-end electron microscope with a cryogenic stage. Relative to a microscope used for conventional SPA, several hardware and software modifications are required to collect low-noise spectral images. EELS spectra have to be collected pixel-by-pixel in STEM mode, requiring the respective alignments and scanning functionality. We found an annular dark-field (ADF) detector useful for faster navigation and focusing. An energy spectrometer with an energy resolution of at least 1 eV and stability over several days is desirable. The detector should fulfill several requirements: the readout noise should be low to allow accurate detection of single electron events. This is especially important for the low-dose acquisitions used in this method, where there are only a few electron events in the core-loss region of a given spectrum. The dynamic range should be large so that the intense zero-loss peak can be recorded simultaneously. Additionally, high frame rates are required for reasonable acquisition speeds, as a detector readout is required at every scanning position and, therefore, 16 million times per 4k × 4k spectral image, making the acquisition slow. All these requirements are met by modern hybrid pixel detectors. We used a Titan Krios G3 in STEM mode, equipped with conventional ADF detectors, a CEOS Energy Filtering and Imaging Device (CEFID) energy filter[19] and a postfilter Dectris ELA hybrid pixel detector[20] for spectral image acquisition. In an attempt to balance detector performance with aberration effects, we chose an acceleration voltage of 200 kV for our experiments. Different energy ranges of an exemplary spectral image recorded in this way are shown in Extended Data Fig. 1.

## Establishing an automated workflow for data collection

REEL analysis depends on large data sets to accumulate sufficient signal for elemental detection, which is substantially facilitated by automated acquisition of spectral images (Extended Data Fig. 2). For this, we use a combination of the microscope automation software SerialEM and the filter-control software Panta Rhei. SerialEM is used for setting up and navigating acquisition positions and for automatically focusing on a carbon area adjacent to the target position. SerialEM then calls an external Python script that triggers spectral image acquisition and saving in Panta Rhei via its remote procedure call (RPC) interface. After the acquisition, the stage is moved to the next position, and autofocus is carried out by SerialEM, while, in parallel, Panta Rhei is assembling and saving the previous spectral image. In this way, the acquisition can run for several days, producing one 4k × 4k spectral image every 23 min. The speed of the data collection is limited by the pixel dwell time, which depends on the maximum frame rate of the detector and by the assembly and saving time for a spectral image in Panta Rhei. We acquired the spectral images with a dwell time of 30 μs (~33 kHz), which is beyond the specification for the Dectris ELA, but we did not observe data loss in this mode. We found that the addition of gold fiducials to the carbon film aids the autofocus routine, which is, in our experience, much less robust in STEM than in conventional transmission electron microscopy (TEM).

## Determining particle poses from EBF images

To calculate particle poses, a reference image is required, which could be derived in a number of ways. A conventional STEM detector records amplitude contrast signals due to differences in scattering angle, with elastic scattering angles being typically higher than inelastic scattering angles. The signal of elastically scattered electrons can be collected

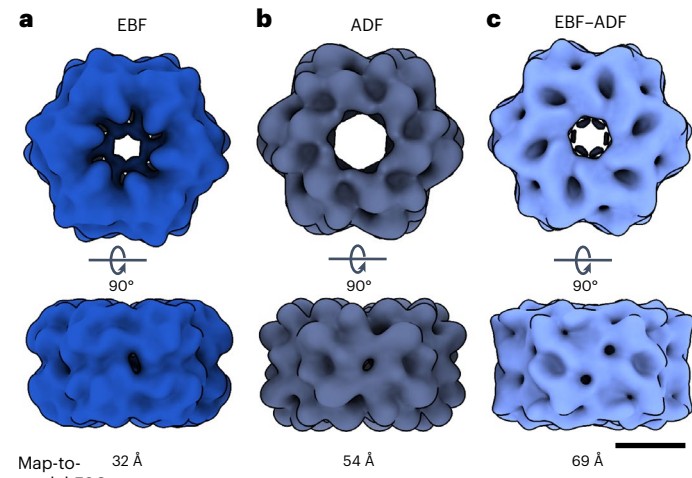

**Fig. 2 | Evaluating the use of different imaging modalities for the refinement of reconstruction poses. a**, The reconstruction refined from EBF images, which were extracted from the zero-loss region of spectral images. **b**, The reconstruction refined from ADF images. **c**, The reconstruction refined from combined images, generated by subtracting the ADF images from the EBF images. The volume shown in **a** resolves the most features of worm hemoglobin, showing a better-quality reconstruction than **b** or **c** and a higher-resolution map-to-model FSC (PDB-5M3L)[25] at a 0.5 cutoff. The SPA reconstructions of WH were refined from the same subset of 3,086 particles, extracted from the different images. The scale bar corresponds to 10 nm.

either by recording their presence at higher scattering angles, forming an ADF image, or their absence at lower angles, forming a bright-field image. In our experiments, the camera length was adjusted so that the full bright-field disk entered the entrance aperture of the energy filter, excluding (mostly elastically scattered) electrons scattered at higher angles (convergence angle $\alpha$ = 6 mrad, collection angle $\beta$ = 8 mrad). Summing the counts in the zero-loss portion of the spectral image, therefore, gives an elastic bright-field (EBF) image.

It is possible to collect an ADF image in parallel to spectral image acquisition. For a small data set of 3,086 particles of worm hemoglobin (erythrocruorin) from *Lumbricus terrestris*, we compared the quality of reconstructions, aligned with a RELION[21] Class3D job, from simultaneously collected dark-field and bright-field images. The collection angle range for the dark-field detector was 15–150 mrad. We also compared a reconstruction from images for which the ADF images were subtracted from the EBF images (Fig. 2). The bright-field images performed best, resolving most features in the reconstruction. The dark-field and mixed reconstructions, in comparison, appear more noise dominated. This is most probably a consequence of larger amounts of detector noise in the ADF than in the EBF images. For the larger data sets, we, therefore, did not acquire simultaneous dark-field images and used only the EBF images as reference.

From a data set of 1,142 spectral images of rabbit ryanodine receptor 1 (RyR1), we extracted EBF images from an energy region of ±3.6 eV. From these, we picked 460,651 particles with a manually trained model in crYOLO[22], which we extracted and subjected to 2D classification for cleaning, 3D classification for prealignment and refinement in RELION[21], using an initial model generated from the data, lowpass filtered to 60 Å, as reference. No contrast transfer function correction was applied during processing. The refinement yielded a reconstruction at 24 Å resolution after postprocessing (Fig. 3 and Extended Data Table 1). The same procedure was performed on a smaller data set of worm hemoglobin (WH). For this sample, 16,648 EBF particle images yielded a resolution of 27 Å after postprocessing (Extended Data Fig. 3 and Extended Data Table 1). These refinements provided particle positions and poses to allow reconstruction of energy-loss data.

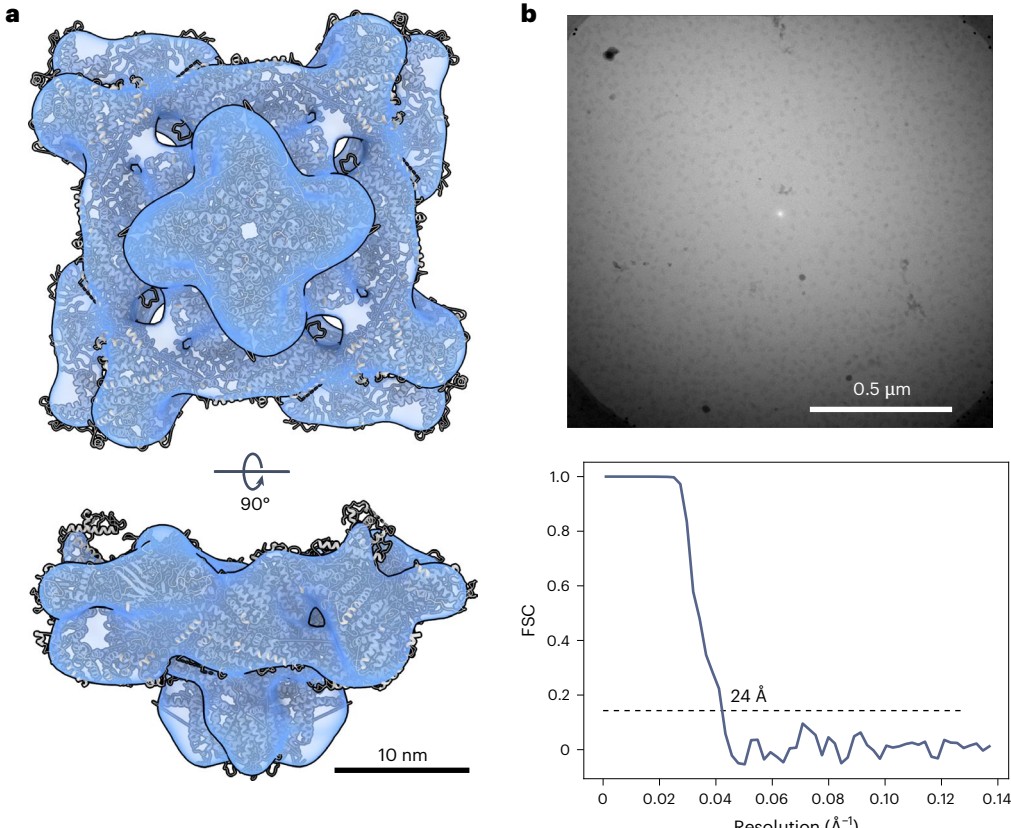

**Fig. 3 | Reference reconstruction of RyR1. a**, Refined reconstruction of RyR1 from a data set of 460,651 particle images, which were extracted from the EBF images generated from zero-loss peak counts. The map shows a good fit—albeit at low resolution—to the published model of rabbit RyR1, PDB-5TAQ ref. 24, which is displayed as cartoon model. The refined poses can be used for reconstruction of energy-loss volumes. **b**, An exemplary extracted EBF micrograph. **c**, The gold-standard FSC for this reconstruction has a 0.143 cutoff at 24 Å.

## Reconstruction of energy-loss data to generate 3D EELS maps

A schematic of the reconstruction procedure is shown in Extended Data Fig. 4. The star files from the reference refinements were manipulated to point to energy-loss images for each 0.77 eV energy bin of the spectral images. For each energy bin, particles were extracted and reconstructed to a 3D volume, using the command relion_reconstruct. To maintain interpretable EELS spectra after reconstruction, the volumes should only undergo linear transforms so that an external intensity scale is maintained. These terms are not met by relion_reconstruct, as this function sets the Fourier origin pixel of each image to zero, thereby roughly normalizing the intensity of the reconstruction. We compiled a version of RELION without this normalization and used this version for reconstruction. This modification comes at a cost of artifactual density, which is strongest at the corners of the reconstructed box (Extended Data Fig. 5). This effect can be minimized by oversampling the Fourier volume using the pad option. We found that a padding factor of 8 renders the intensity of the artifact sufficiently low for it not to be visible at threshold levels used for inspecting the reconstructions, although this resulted in noisier spectra. For this reason, all spectra displayed in this work are from reconstructions performed with two-fold zero-padding and all displayed volumes from reconstructions performed with eightfold zero-padding for better visualization. The modified reconstruction procedure maintains the intensity profile of the EELS spectrum of the constituent particle images (Fig. 4).

Analyzing the spectrum summed across all voxels of the RyR1 reconstructions, ionization edges for the most abundant elements are clearly visible: the K edges of carbon at 284 eV, oxygen at 532 eV (with a small prepeak indicating the expected presence of gaseous oxygen due to radiation damage[23]) and nitrogen at 401 eV, as well as

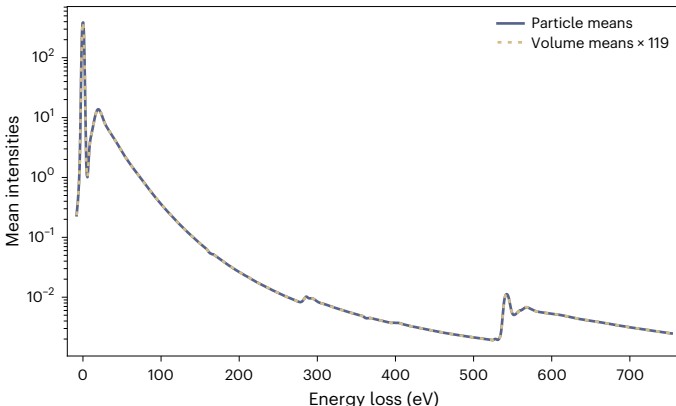

**Fig. 4 | The mean intensities of particle images and respective 3D reconstructions maintain the same profile along the energy-loss spectrum.** The mean intensities of all voxels in all particle images at each energy loss are indicated with the continuous blue line and mean intensities of the volumes, reconstructed from these images with a modified version of relion_reconstruct, are indicated with the yellow dashed line. Applying an empirical constant scaling factor of 119 to the mean intensities of the reconstructions, the curves become nearly identical, showing that the EELS profile is maintained by the reconstruction procedure.

the $L_{2,3}$ edges of the buffer components chlorine at 200 eV, potassium at 294 eV and phosphorus at 132 eV (Fig. 5a,b). Supplementary Movie 1 shows a sequence of all energy-loss reconstructions. There are evident changes in the 3D localization of the signal with changing energy loss.

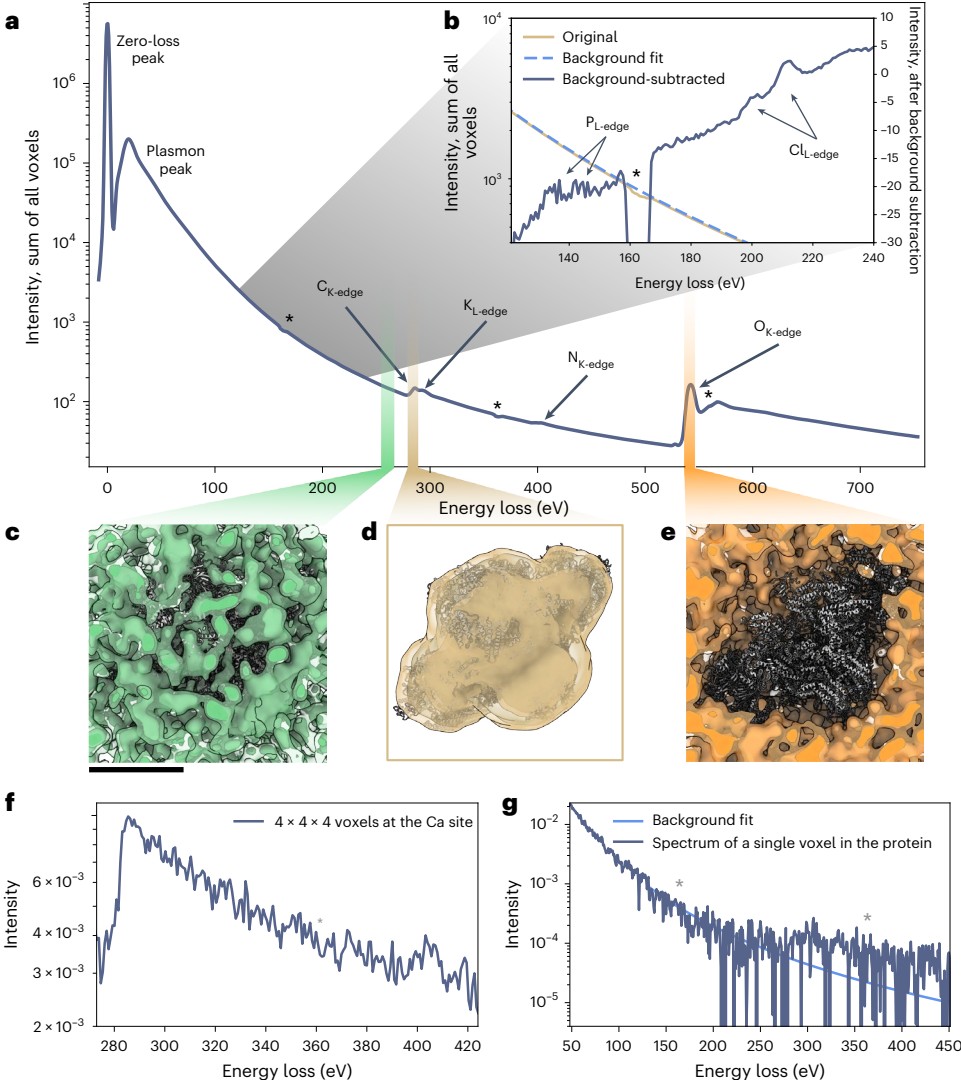

**Fig. 5 | Selected spectra and volumes of reconstructed EELS data of RyR1.**
**a**, The spectrum of the sum of all intensities in the EELS reconstructions shows elemental edges for the most abundant elements: carbon, potassium, nitrogen and oxygen. The asterisks mark regions where the spectra show dips due to double-wide pixels at the tile boundaries of the detector, which are not perfectly accounted for by the detector's flatfield correction. **b**, For the region between 121 and 240 eV, a background subtraction was performed, based on a fit of the spectrum between 121 eV and 127 eV. The subtracted spectrum shows additional edges for phosphorus and chlorine. **c–e**, Gaussian-filtered reconstructions for three 4.6 eV regions of the spectrum are shown together with an atomic model of rabbit RyR1, PDB-5TAQ ref. 24. Before the carbon edge (**c**), the reconstruction

shows a uniform distribution of intensity between the protein and the solvent. At the carbon edge (**d**), the reconstruction shows colocalization with the protein and micelle, and at the oxygen edge (**e**), the reconstruction shows colocalization with the solvent. The reconstructions reflect the known local distributions of carbon and oxygen. **f**, The spectrum of summed intensities of a 4 × 4 × 4 voxel region at the known calcium-binding site does not show a calcium edge discernible above noise at 346, 350 eV. **g**, The spectrum of a single voxel within the protein shows a clearly discernible carbon edge. A background fit based on the region 120–240 eV is displayed for better visualization. The scale bar near **c** corresponds to 10 nm and applies to **c–e**.

These changes coincide with the major features in the spectrum; in the core-loss region before the carbon edge (237.5–242.1 eV), the intensity shows no apparent correlation with the complex. In contrast, at the carbon edge (283.2–287.8 eV), the density colocalizes with the protein and the micelle. At the oxygen edge (540.4–545.0 eV), the density shows an inverse correlation with the complex, in line with the higher oxygen concentration in the solvent (Fig. 5c–e). These differences reflect the known elemental distributions of the complex. In addition to elemental features, analysis of the low-loss region (energy loss, $\Delta E \leq 50$ eV) of the spectrum shows differing spatial distributions for different energy windows of the plasmon peak (Extended Data Fig. 6).

In principle, EELS analysis can detect all elements other than hydrogen (except as $H_2$). Of the known elements in our sample, we detect all except hydrogen, zinc, sulfur and calcium. As the energy

range and spectrum positioning on the detector were optimized for other edges in this data set, we do not expect to detect zinc because its major absorption edge (1,020 eV, 1,043 eV) lies beyond our chosen energy range, or sulfur (165 eV) because the associated energy loss coincides with an artifact at the detector tile edge (Fig. 5b).

The spectrum of a 4 × 4 × 4-voxel region near the known calcium-binding site formed by residues including E3893 and E3967, and T5001 (ref. 24) does not show a calcium $L_{2,3}$-edge signal discernible above noise, indicating that we have not reached single-atom sensitivity with this data set (Fig. 5f). The spectrum of a single voxel in the protein, which contains on average 1.7 atoms of carbon, clearly shows a carbon edge (Fig. 5g), giving an indication that we are approaching the range of single-atom sensitivity. An important difference to the detection of calcium at its binding site is that the carbon signal is less

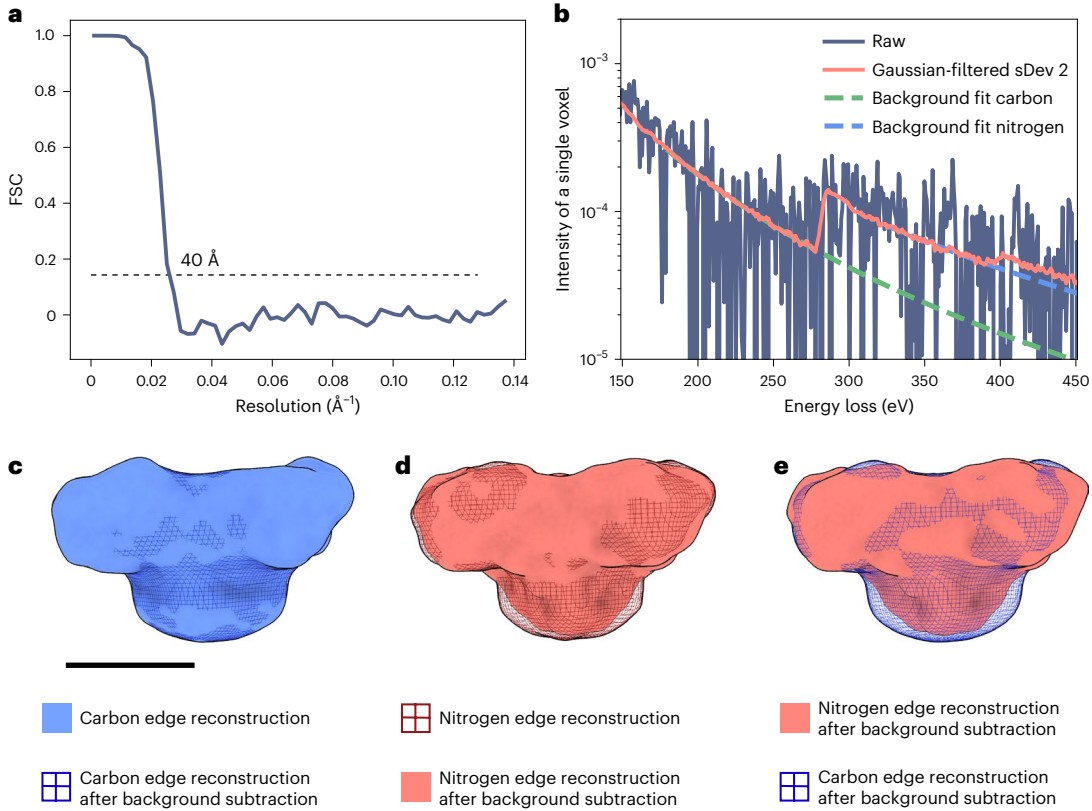

**Fig. 6 | Evaluating the resolution of EELS reconstructions and the feasibility and quality of localized background subtraction. a**, The resolution of the energy-loss reconstruction at the carbon edge is estimated by reconstructing and summing halfmaps in the edge region and calculating their FSC, giving a resolution of 40 Å. **b**, A spectrum of a single voxel of raw reconstructions, a spectrum of the same voxel after applying Gaussian filtering with a standard deviation (sDev) of 2 voxels to the volumes and background fits to the latter before the carbon and before the nitrogen edge. **c**, The carbon reconstruction looks similar before and after background subtraction. **d**, The nitrogen reconstruction after background subtraction shows less intensity in the region of the micelle, as the background subtraction effectively removes the signal due to the long tail of the carbon edge below the nitrogen edge signal. Comparing the background-subtracted maps for carbon and nitrogen, the differences in the micelle region are reflective of the lower N:C ratio in the micelle than in the protein region. The scale bar near **c** corresponds to 10 nm and applies to **c**–**e**.

affected by inaccuracies in particle poses, as surrounding voxels also contain carbon.

Spectral reconstructions of WH were produced in the same way as for RyR1.

The spectrum of the sum of all voxel values of the reconstructions similarly shows carbon, oxygen and nitrogen K edges, as well as $L_{2,3}$ edges for chlorine and potassium (Extended Data Fig. 7). Supplementary Movie 2 shows the full series of energy-loss reconstructions.

### Resolution of the RyR1 reconstruction at the carbon edge

The resolution of reconstructed energy-loss volumes can be estimated by halfmap Fourier shell correlation (FSC) analysis. For the sum of spectral reconstructions at the carbon edge (283.2–287.8 eV), the FSC between the sums of reconstructed halfmaps was calculated (Fig. 6a). As the halves were separated according to the gold standard during refinement, we used a correlation cutoff of 0.143, yielding a resolution of 40 Å.

### Per-voxel background subtraction

To obtain an element-specific signal by EELS analysis, it is necessary to account for the background signal. In the case of carbon, the localization of the signal with the protein in the spectral reconstructions has a clear onset at the carbon edge, showing that the elemental signal is strong enough to dominate above the background at this edge (Fig. 5c,d and Supplementary Movies 1 and 2). For smaller edges, an element-specific signal can only be visualized after accounting for

the signal of the decaying background in the EELS spectra. For our 4D spectral volumes, this requires per-voxel background subtraction at the relevant edge(s).

To assess whether such a background subtraction is feasible with our data, we performed per-voxel background subtraction at the carbon and nitrogen K edges. Individual voxel spectra show a relatively high level of noise, especially at larger energy losses (Fig. 6b). To ensure sufficient signal for accurate background modeling, we applied Gaussian filtering with a standard deviation of 2 voxels to the reconstructed volumes (Fig. 6b). For each voxel, we then fit the region between 160 eV and 250 eV for carbon and between 325 eV and 375 eV for nitrogen with a power law function (Fig. 6b) and subtracted the extrapolated fit from the subsequent energy bins. The true elemental volumes were then generated by summing the background-subtracted values in the edge regions of carbon (283.2–287.8 eV) and nitrogen (403.3–407.9 eV). For the carbon edge, the background-subtracted volume appears very similar to the nonsubtracted summed volume (Fig. 6c). A larger difference is apparent for the maps at the nitrogen edge (Fig. 6d). The nitrogen K edge lies on the long tail of the carbon K edge, so that the reconstructed volumes at the nitrogen edge also contain scattering contributions from carbon atoms. Comparing the unsubtracted and background-subtracted reconstructions, the reconstruction without background subtraction shows stronger density in the region of the micelle than the map with background subtraction, when adjusting to similar thresholds for the protein (Fig. 6e). This reflects the higher carbon:nitrogen ratio of the micelle

in comparison with the protein. The micelle is composed mainly of 3-[(3-cholamidopropyl)dimethylammonio]-1-propanesulfonate (CHAPS), for which the ratio is 16:1, while it is 3.6:1 for RyR1.

This result demonstrates the importance and feasibility of performing background subtraction on REEL volumes, to derive specific elemental distributions from reconstructed energy-loss volumes.

## Discussion

To fully understand macromolecular complexes at an atomic level, a technique for elemental mapping is needed. This method should ideally provide well-resolved 3D localization and be compatible with low-dose imaging of cryo-preserved samples while providing single-atom sensitivity.

Here, we developed a workflow for a new technique, REEL analysis, in which EELS data are reconstructed in three dimensions, allowing a high total dose to be accumulated from many individual spectral images of a cryo-preserved complex, each collected at less than $100\ e^-\ Å^{-2}$. We present proof-of-principle data, showing that we can use poses determined from a reference refinement to reconstruct a 4D data set: a full energy-loss spectrum of 3D volumes, which correctly reproduces known distributions of abundant elements, including light elements that are inaccessible to anomalous scattering techniques. A 3D per-voxel background subtraction can be performed on this 4D data set to disentangle elemental signal from background.

We have compared the use of ADF and EBF images, as well as a difference image of the two, for determination of reconstruction poses. We found the elastic bright field to give the best results, which may be related to differences in noise between the detectors used in our setup. To facilitate the collection of large data sets, we have established a workflow for automated data collection. We have further established an image processing pipeline for spectral reconstruction, using Python-based tools and an adapted version of the reconstruct function of RELION, which maintains an external intensity scale between reconstructions.

The resolution of the elemental maps for the RyR1 data set can be estimated to around 4 nm on the basis of halfmap reconstructions at the carbon edge. Although this resolution does not allow neighboring atoms to be distinguished, it is sufficient for localization on the order of domains. In combination with a high-resolution reconstruction from TEM, even a low-resolution elemental map may be sufficient to assign unknown densities.

The results presented here fall short of single-atom sensitivity, which would be desirable for most applications of the method. With 460,651 fourfold symmetric particles, collected at a dose of $92\ e^-\ Å^{-2}$, the effective accumulated dose for the RyR1 reconstruction is around $1.7 \times 10^8\ e^-\ Å^{-2}$, which exceeds a previously estimated minimum dose of $10^7\ e^-/Å^{-2}$ for single-atom sensitivity[3] by more than an order of magnitude. It is important to note that this estimate was performed on the basis of dry protein on carbon foil (thickness 4 nm) for which less background is expected than for our ice-embedded samples (thickness >30 nm), correspondingly increasing the required total dose to achieve a similar SNR. Moreover, the estimate referred to elemental detection in a single exposure. For single-particle averaging techniques, imperfect alignment accuracy increases the required cumulative dose. To distinguish between the effects of cumulative dose and the effects of alignment accuracy on sensitivity, an important observation is that, for a single voxel in the protein, containing one to two carbon atoms, the corresponding spectrum features a discernible carbon K edge. The carbon signal is not strongly affected by alignment, since neighboring voxels in the protein also contain carbon; therefore, the detection of carbon but not less abundant elements in single-voxel spectra points to an important role of alignment inaccuracies in limiting the current data set.

Both image quality and processing algorithms influence the accuracy of the alignment. Image resolution in STEM depends crucially on the size of the imaging probe. Over the course of overnight data collection on our microscope, during which alignments are not retuned, the probe size increased substantially due to instabilities in direct alignments. This probably limits the attainable resolution for most of the images. Moreover, our reference images are formed by amplitude contrast, which is less efficient than phase contrast for SPA samples, which are generally thin and composed of light elements. This limits not only the achievable alignment accuracy but is also probable to pose limitations to the study of small complexes. We circumvented this limitation for this initial study by choosing large complexes (around 2.3 MDa for RyR1 and 2.8 MDa for WH[25]). STEM phase retrieval methods are an active area of investigation[26,27] and hold potential for improvements in resolution and applicability to small complexes. A combination of these methods with EELS is constrained by practical considerations but holds promise for the future[28–31].

Processing algorithms used in this work are standard SPA algorithms and, therefore, are optimized for TEM phase contrast images. Adjustments for amplitude contrast images hold potential for improvements.

For this study, we chose a dose of $92\ e^-\ Å^{-2}$ to balance the high-dose requirements of EELS measurements with the cumulative effects of radiation damage. This exceeds typical doses used in SPA, which are often limited to $30–60\ e^-\ Å^{-2}$, but is below the doses at which bubble formation causes larger-scale disruption[32]. Future studies could investigate the use of dose weighting[33,34] to mitigate the effects of radiation damage on resolution, although a study in TEM mode suggests that our resolution is not limited by the effects of radiation damage[34].

Regardless of possible improvements in image quality, an increase of data quantity will certainly contribute to improved elemental sensitivity. For this, an increase in data collection speed is desirable, considering that this study already required several weeks of instrument time. We, therefore, require faster detectors, a development that is already underway[35–37]. Faster detectors would further allow dose fractionation and motion correction, which are currently not practical due to limitations in detector frame rate.

Reaching single-atom sensitivity would enable a deeper understanding of macromolecular complexes than is currently possible by any technique. In addition to metal ions, identification of lighter ionic species has the potential to enrich our understanding of ion channels and transporters, as well as many complexes essential to transcription and translation. Detection of sulfur or selenium substituents has the potential to guide chain tracing in poorly ordered protein regions. Mapping of lipid interactions, especially in combination with elementally labeled lipids, has the potential to enable understanding of not only stable but also transient interactions that are essential for membrane protein function.

The data presented here provide proof-of-principle for the method of REEL analysis. They indicate that increased image quality and data set size are the most important avenues for further improvements in resolution and sensitivity. With future developments, REEL analysis has the potential to enrich our understanding of the structure and function of macromolecular complexes.

## Online content

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

## Methods

### Sample preparation

Rabbit RyR1 was purified as described previously[38] with modifications. All steps were performed at 4 °C. In brief, 200 g of frozen rabbit skeletal muscle tissues (Pel-Freez Biologicals) were blended for 120 s in 2 × 400 ml of buffer A (20 mM Tris−maleate pH 6.8, 10% sucrose, 1 mM dithiothreitol, 1 mM ethylenediaminetetraacetic acid (EDTA), 0.2 mM phenylmethylsulfonyl fluoride (PMSF), 1 mM benzamidine) and centrifuged at 3,000*g* for 10 min. The supernatant was filtered through a cheesecloth and ultracentrifuged at 200,000*g* for 45 min. The membrane pellet was collected, flash frozen in liquid nitrogen and stored at −70 °C for later use. The membrane fraction after thawing was solubilized in buffer B (50 mM 2-[4-(2-hydroxyethyl) piperazin-1-yl]ethanesulfonic acid (HEPES) pH 7.5, 1 M NaCl, 1 mM ethylene glycol-bis(β-aminoethyl ether)-N,N,N′,N′-tetraacetic acid (EGTA), 2 mM tris(2-carboxyethyl)phosphine (TCEP), 0.1 mM PMSF, 1% CHAPS, 0.2% soybean L-α-phosphatidylcholine, 1:1,000 diluted protease inhibitor cocktail (PIC; Millipore Sigma 539134)) for 1 h. Then, an equal volume of buffer C (buffer B lacking 1 M NaCl) and His-GST-FKBP12.6 (10 mg) were added, and this was stirred for another hour. The solubilized mixture was ultracentrifuged at 200,000*g* for 45 min. The supernatant was mixed with 3 ml of glutathione sepharose 4B resin (Cytiva) and stirred for 1 h. The resin was poured into a gravity column, washed with buffer C (50 mM HEPES pH 7.5, 0.5 M NaCl, 2 mM EGTA, 2 mM TCEP, 0.1 mM PMSF, 0.5% CHAPS, 0.1% soybean L-α-phosphatidylcholine, 1:1,000 PIC) and eluted with buffer C plus 20 mM glutathione. The His-GST tag was removed by overnight incubation with tobacco etch virus protease (1 mg). The cleavage product was concentrated and applied to a Superose 6 Increase 10/300 GL column (Cytiva) in buffer D (20 mM HEPES−KOH pH 7.5, 250 mM KCl, 40 µM free $Ca^{2+}$ buffered with 2 mM EGTA, 0.375% CHAPS, 0.001% 1,2-dioleoyl-sn-glycero-3-phosphocholine (DOPC), 2 mM TCEP, 0.1 mM PMSF, 1:1,000 PIC). The peak fractions containing RyR1 were combined and concentrated to -15 mg ml$^{-1}$ (measured by NanoDrop using the A280 method). The RyR1-FKBP12.6 sample was mixed with calmodulin (equilibrated in buffer D) to a final concentration of 12 mg ml$^{-1}$ and 100 µM, respectively. Holey carbon grids (Quantifoil Cu 300 mesh, R2/2) were prepared with fiducials by glow-discharging, adding 3 µl of 10 nm protein A immunogold solution and blotting from the back. The grids were left to dry overnight in a desiccator. A total of 2.5 µl of the RyR1-FKBP12.6-calmodulin sample were applied and blotted for 3 s (blot force of 6–9) using ashless blotting paper, and the grids were subsequently plunge frozen in liquid ethane using a Vitrobot Mark IV (Thermo Fisher Scientific) at 4 °C and 100% humidity.

For the WH sample, earth worms (Canadian Night Crawler) were purchased from a fishing supplier. The blood was extracted from the seventh segment and diluted with saline solution of 50 mM NaCl and 10 mM $MgCl_2$. C-flat grids with a hole size of 1.2 µm and a hole spacing of 1.3 µm were used for this sample. The grids were prepared with fiducials by glow-discharging, adding 3 µl of 10 nm protein A immunogold solution to the carbon and blotting from the back. The grids were left to dry over a minimum of one night before they were used for sample preparation. They were then again glow-discharged and 3 µl of sample were applied, before blotting and plunge freezing in liquid ethane in a Vitrobot Mark IV, using a blot force 0 and 8 s blot time at 4 °C and 100% humidity.

### Spectral image data collection

The data were recorded on a Titan Krios G3 with STEM capability, equipped with high-angle annular dark-field (HAADF) (Fishione) and ADF (Thermo Fisher Scientific) detectors, a CEFID energy filter[19] and a postfilter Dectris ELA hybrid pixel detector[20]. For the acquisition of spectral images, scan generation, beam blanking and data compression were performed by the CEOS software Panta Rhei, with scanning controlled through an external scan generator (TVIPS). For acquisition of overview, alignment and focus images, the Thermo Fisher Scientific

scan generation and beam blanking were used. The experiments were performed at an acceleration voltage of 200 kV, convergence angle of 6 mrad, spot size 10, gun lens 3, 50 µm C2 aperture and a magnification of ×80,000, corresponding to a pixel size of 3.65 Å. The collection angle for electrons entering the energy filter corresponded to 8 mrad. This was limited by the hole in the dark-field detector, which was inserted during acquisition, as this was smaller than the entrance aperture of the filter. To determine the dose rate, we first calibrated the average number of pixels activated per electron hit by looking at a sparse region of a spectrum, where single electron hits were clearly discernible as clusters of activated pixels. We then divided the total electron counts per time in an image of the beam over vacuum by this factor and the pixel area. This is enabled by the linear count response of the detector to electrons up to $5 × 10^6$ counts per pixel per second, which was not exceeded in the experiment [20]. The pixel size was calibrated by maximizing the 0.5 FSC between an extracted bright-field reconstruction and a TEM reconstruction with a carefully calibrated voxel size of the same sample. For maximum frame rates of the detector of up to 33,333 Hz, only a 66 × 1,024 pixel area region of interest of the detector was read out. The recorded spectra were then summed in the nondispersive direction by Panta Rhei before saving of spectral images in Hyperspy format with lossless compression. The energy filter was set to disperse 770 eV (from −25.4 eV to 745.6 eV) over the longer dimension of the detector, yielding energy bins of 0.75 eV. The bin size was later adjusted by a factor of 532/515 on the basis of the position of the oxygen edge relative to the zero-loss peak. STEM direct alignments were performed manually on an area of carbon film adjacent to the acquisition area, immediately before beginning data acquisition. Filter alignments were performed over vacuum using the automated tuning routine provided in Panta Rhei.

The acquisition was automated using SerialEM[39] for navigation and autofocus and using Panta Rhei for spectral image acquisition and saving. In SerialEM, the overview images of the entire grids were acquired in TEM mode, as stitched maps at low magnification. Suitable squares were then selected and recorded at an intermediate magnification in STEM mode, on the DF2 darkfield detector. Intact holes were selected as acquisition positions on the grid square maps. An automated acquisition routine was then triggered, which functions as follows: at each acquisition position, a SerialEM script moves the stage to the coordinates, and the hole is further aligned on the basis of a low-dose image from the DF2 detector. The STEM autofocus routine is then run on an adjacent area of carbon, and an external RPC script is triggered. SerialEM relies on the internal scan generator of the microscope (Thermo Fisher). The RPC script calls functions for spectral image acquisition and saving in Panta Rhei. This requires a switch to the external (TVIPS) scan generator, which is automatically performed by Panta Rhei. The files are saved in hyperspy format. After a wait time of 12 min, when the current spectral image acquisition has been completed, but data are still being processed and saved by Panta Rhei, the SerialEM script resumes and moves the stage to the next imaging position where the autofocus routine is again started. A wait time is set to ensure a minimum of 23 min between one call of the RPC script and the next to ensure that the previous file is saved before starting a new acquisition.

The 1,142 4k × 4k × 1k spectral images of RyR1 and 61 spectral images of WH, of which 7 included a simultaneous dark-field image, were collected with a dwell time of 30 µs per pixel at a calibrated total dose of 92 e$^-$ Å$^{-2}$. The collection angle for the dark-field detector was 15–150 mrad.

### Reference reconstructions for pose determination

Using the Hyperspy framework[40] (version 1.6.4), the mean zero-loss peak position was calculated for every pixel in a set of sample spectral images by determining the location of maximum intensity. We found that the position was stable within the instrumental limits on energy resolution throughout the spectral images (standard deviation of 0.8 eV; full-width at half maximum of the zero-loss peak in a single high-dose

spectrum 2.2 eV), so the mean position for each spectral image was calculated from a central strip of pixels (40 × 4,096) perpendicular to the scanning direction for the correction of the zero-loss peak position. EBFs were extracted as tiff files from an energy range of ±3.5 eV around the zero-loss peak center. This encompasses 99% of the peak. Within these micrographs, particle positions were determined using crYOLO[22] with a manually trained model. The particles were extracted and further processed in RELION-4[21]. No motion correction, dose weighting or contrast transfer function correction was applied. A 2D classification was applied for additional data set cleaning. A total of 460,651 particles were selected for the RyR1 data set and 16,648 particles for the WH data set. The selected particles were prealigned using a Class3D job with a *T* value of 1 and 40 iterations, using as reference a 60 Å lowpass-filtered initial model, generated from the data for each data set, and then subjected to 3D refinement, using a soft solvent mask. C4 symmetry was applied for RyR1 and D6 symmetry for WH. The refinement resolutions were determined by gold-standard FSC calculation in a PostProcess job.

For the spectral images for which a simultaneous dark-field image was collected, 3,086 particles were picked on EBF images, generated as above. An additional set of images was produced by subtracting the dark-field images from the corresponding EBF images. The particles were extracted from all three sets of images and independently subjected to alignment by 3D classification with 40 iterations. For each map, an FSC to a map on the basis of PDB-5M3L (ref. 25), generated by ChimeraX's[41] molmap command, was calculated and resolutions were estimated from a 0.5 cutoff.

### Reconstruction of energy-loss data

In a second Hyperspy script, energy-loss micrographs were extracted as tiff files from the spectral images for both data sets. For each energy bin, all particles were reextracted with the pose information from the Refine3D star file. No normalization was applied to the particle images. For reconstruction, RELION was recompiled without line 714 in src/reconstructor.cpp, which otherwise sets the Fourier origin voxel of the reconstruction box to zero. A volume was then reconstructed for each energy, using the command relion_reconstruct with twofold or eightfold padding and full gridding. A single reconstruction (at 284 eV) was also made without any padding for comparison. For the maps of the pre-edge region, the carbon edge and the oxygen edge, the volumes for 237.5–242.1 eV, 283.2–287.8 eV and 540.4–545.0 eV, respectively, were summed. For display, a Gaussian filter with a standard deviation of 2 voxels was applied. For the RyR1 data set, the halfmaps were reconstructed for the volumes between 283.2 and 287.8 eV by applying the halfmap option in relion_reconstruct, such that the halves were separated in the same way as for the EBF reference reconstructions. The maps were then summed for all of these volumes for each half data set and an FSC curve was calculated between the sums of the halfmaps, using relion_postprocess and a suitable mask. The mean intensities of the particles at each energy were determined from all pixels in all particle images that contributed to each reconstruction of the RyR1 data set. In Fig. 4, they are compared with the mean voxel values of the resulting reconstructions.

Volumes were analyzed using ChimeraX[41], and spectra from different regions of the volumes were analyzed using the Hyperspy[40], mrcfile[42] and NumPy[43] libraries. The plots were created in Python, using the matplotlib library[44]. The position of the expected calcium-binding site for RyR1 in relation to the reconstructed volumes was determined on the basis of PDB-5TAQ ref. 24, which was rigid-body-fit into the EBF reference reconstruction. The average number of carbon atoms in a single voxel of RyR1 was determined by counting all atoms in a sphere located in a protein-dense portion of the RyR1 atomic model (PDB-5TAQ ref. 24), dividing by the volume of the sphere and multiplying by the volume of a voxel in the reconstruction.

Movies were produced in ChimeraX by morphing between the series of reconstructions, Gaussian-filtered with a standard deviation of 7.3 Å, using one frame per reconstruction. The option 'constant volume' was set to 'true'. The spectrum and slider were added in the program Adobe Premier Pro.

### Background subtraction

For background subtraction, the spectral volumes were processed with Hyperspy. The volumes were first individually Gaussian filtered with a standard deviation of 2 voxels, using SciPy's ndimage Gaussian filter[45] and then combined into a 4D EELS object. Background fitting was performed with a power law function and the fast-fitting option disabled in the region of 186–273 eV for carbon and 335–387 eV for nitrogen. The background-subtracted volumes at each energy were then individually saved as mrc files. For the carbon map, the subtracted volumes were summed from 283.2–287.8 eV and for nitrogen from 403.3–407.9 eV. For better visualization, Gaussian filtering was again applied to the background-subtracted nitrogen map.

### Statistics and reproducibility

Representative EBF micrographs for the reference workflows of data sets of RyR1 and WH were chosen from data sets of 1,142 micrographs and 61 micrographs, respectively. In Fig. 5g, a representative spectrum of a voxel, which is located in the protein region of the RyR1 reconstruction, is shown. We studied at least 20 blindly chosen voxels in the protein, all of which show the carbon edge in a similar manner. The spectrum is, therefore, representative.

### Reporting summary

Further information on research design is available in the Nature Portfolio Reporting Summary linked to this article.

## Data availability

All maps shown in this work have been deposited to publicly accessible databanks. For the RyR1 data set, the reference reconstruction and corresponding halfmaps, the summed precarbon, carbon, nitrogen and oxygen maps, as well as the background-subtracted carbon and nitrogen maps, and the summed carbon halfmaps were deposited to the Electron Microscopy Data Bank under accession code EMD-19191. The full spectrum of energy-loss reconstructions is available at EDMOND under https://doi.org/10.17617/3.X1R0AQ. For the WH data set, the reference reconstruction and corresponding halfmaps and the summed precarbon, carbon and oxygen maps were deposited to the Electron Microscopy Data Bank under accession code EMD-19190. The full spectrum of energy-loss reconstructions is available at EDMOND under https://doi.org/10.17617/3.AUERWM. The mean intensity of particle images and the mean intensity of reconstructions as a function of energy loss shown in Fig. 4 are available as a source data file. For analysis, the publicly available atomic models PDB-5TAQ (ref. 24) and PDB-5M3L (ref. 25) were used in this work. Source data are provided with this paper.

## Code availability

Scripts used in the acquisition and processing of the data presented are based upon freely available software packages and are included in Supplementary Information.

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

## Acknowledgements

We are very grateful to H. Stark (Max Planck Institute for Multidisciplinary Sciences) for providing infrastructure and support for the project, and to W. Kühlbrandt and M. Beck (Max Planck Institute of Biophysics) for support. We thank J. Gray (Pennsylvania State University) for collecting preliminary data and Sjors Scheres (MRC-LMB) for helpful comments on the RELION code. We thank J. Castillo, Ö. Yildiz and T. Koske for computing support in Frankfurt and Göttingen and the Central Electron Microscopy Facility of the Max Planck Institute of Biophysics for providing infrastructure and support for sample preparation and screening and S. Welsch for helpful discussions. We are grateful to D. Mastronarde (University of Colorado Boulder) for the development and maintenance of the SerialEM package for automated electron microscopy data acquisition. This project has received funding from the Max Planck Society and the European Research Council under the Horizon 2020 research and innovation programme (grant agreement no. 101116848 to B.J.M.).

## Author contributions

B.J.M. initiated and directed the project. O.P.-G., H.V.D.R. and B.J.M. collected, processed and analyzed the data. D.R. assisted in the microscopy. Y.S.C. and F.V.P. purified RyR1 and prepared cryo-EM samples. O.P.-G. and B.J.M. prepared WH samples. D.L., P.K., M.L. and H.M. assisted with the energy filter integration and automated acquisition. O.P.-G. and B.J.M. drafted the manuscript and revised it in consultation with all other authors.

## Funding

## Competing interests

D.L., P.K., M.L. and H.M. are employed by CEOS GmbH, the manufacturer of the CEFID energy filter used in this work. The authors declare no further competing interests.

## Additional information

**Extended data** is available for this paper at https://doi.org/10.1038/s41592-024-02482-5.

**Correspondence and requests for materials** should be addressed to Bonnie J. Murphy.

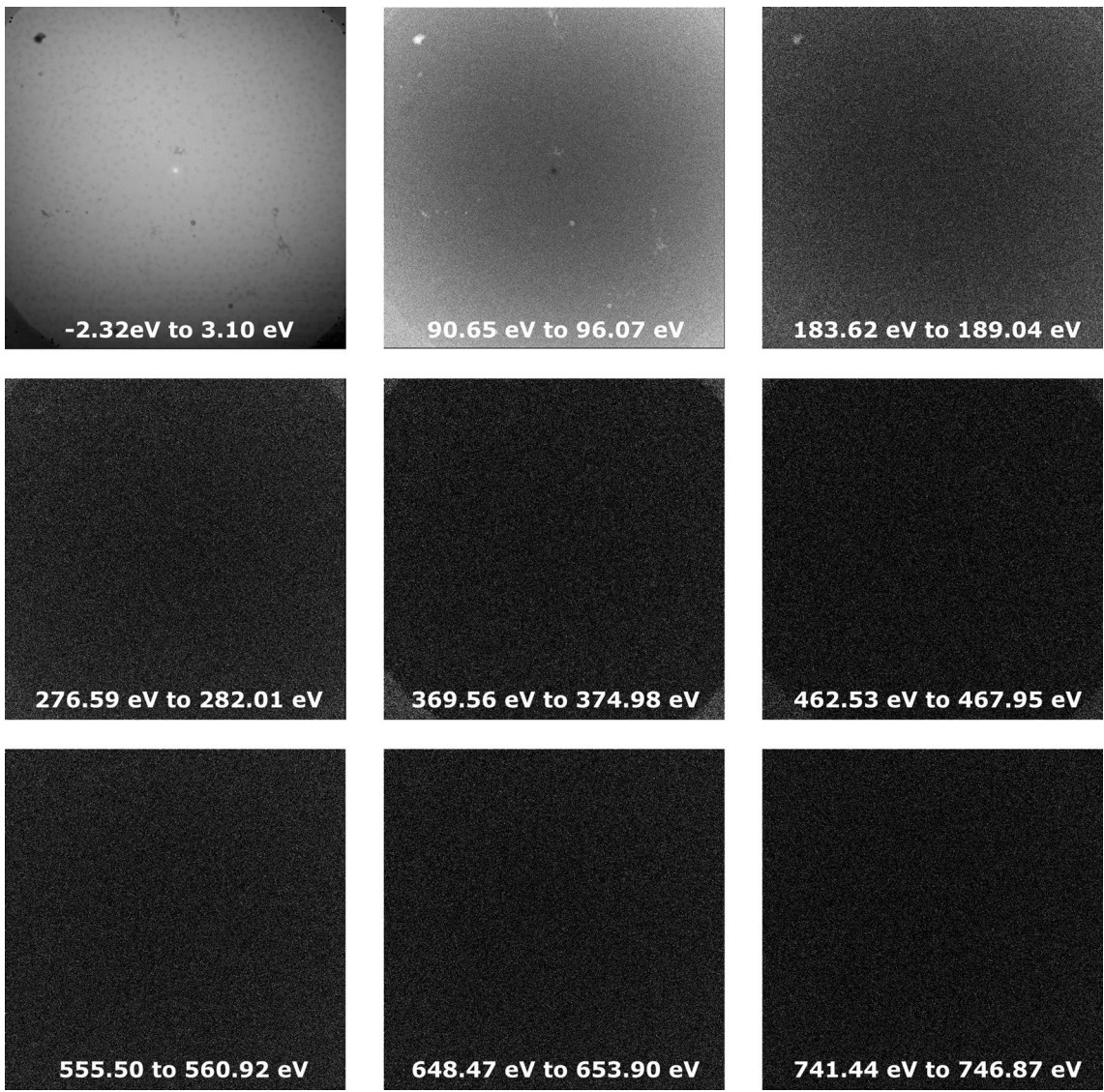

**Extended Data Fig. 1 | Selection of energy-loss layers of an exemplary spectral image.** For better visualisation, sums of seven 0.77 eV-layers are shown in the panels.

**Extended Data Fig. 2 | Schematic of the procedure for automated collection of spectral images.** using the microscope control software SerialEM in communication with the filter software Panta Rhei via its RPC interface. After preparation steps, as indicated, automated acquisition is started. For every acquisition position, the stage and the microscope optics are set for the acquisition by SerialEM, then an external RPC script is called, which triggers acquisition and saving of a spectral image in Panta Rhei.

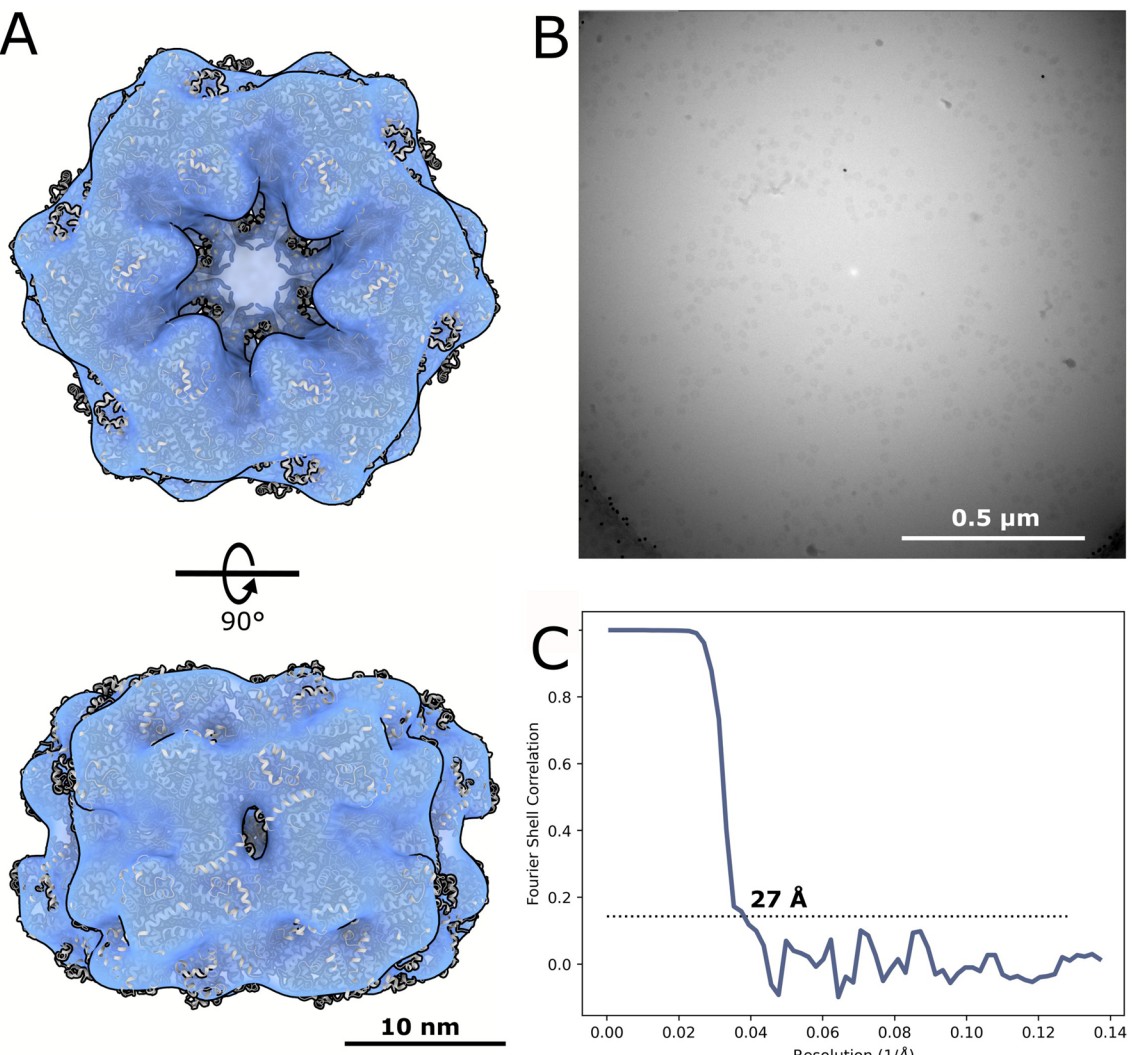

**Extended Data Fig. 3 | Reference reconstruction of WH. (a)** Refined reconstruction of WH from a data set of 16,648 particle images which were extracted from the zero-loss region of spectral images. The map shows a good fit – albeit at low resolution – to the published model of earthworm haemoglobin (PDB-5M3L ref. 25) displayed as cartoon model. **(b)** An exemplary extracted EBF micrograph. **(c)** The gold standard Fourier shell correlation for this reconstruction has a 0.143 cut-off at 27 Å.

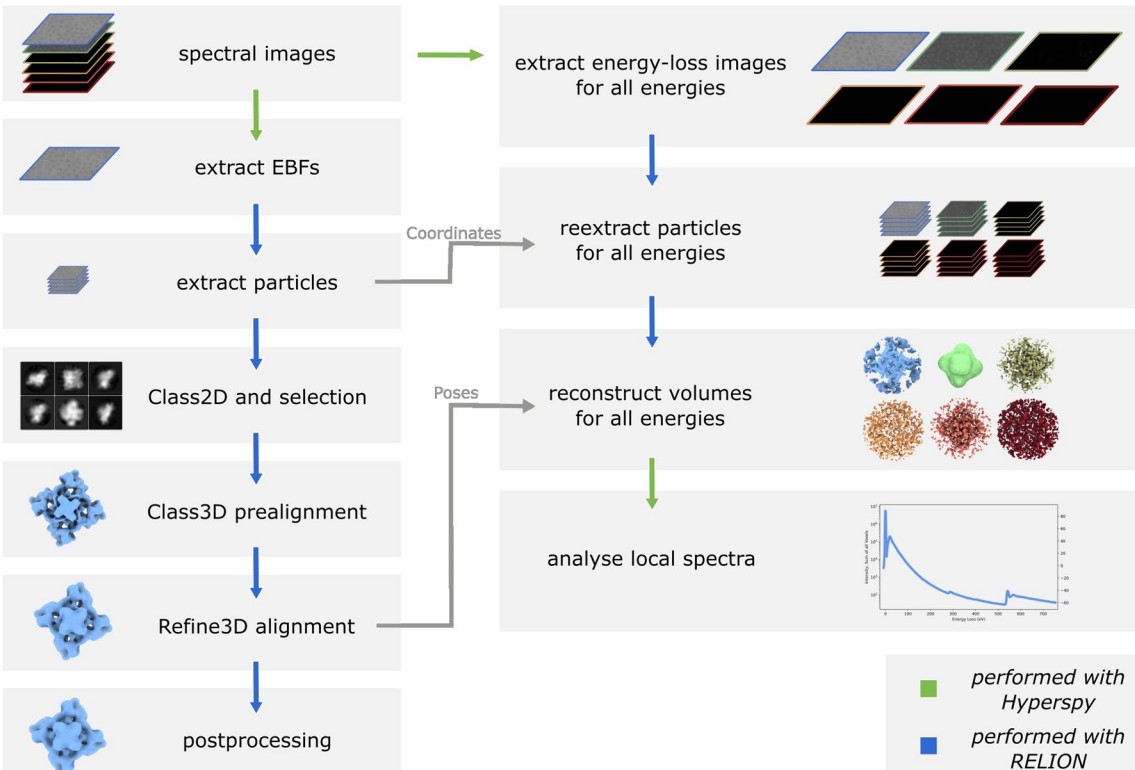

**Extended Data Fig. 4 | Schematic of the data processing procedure.** Spectral images were processed with tools from the Hyperspy framework or the software RELION, as indicated by green or blue arrows, respectively. For the reference workflow, elastic bright-field images were extracted from the spectral images. Suitable particles were picked and extracted from these, subjected to 2D classification for cleaning and then pre-aligned with a 3D classification job and further refined in a 3D Refine job. For resolution estimation, the results were further subjected to postprocessing. For the full spectral workflow, micrographs were extracted from the spectral images at all energies, particles were reextracted from these micrographs using the previously refined particle positions and then reconstructed for each energy on the basis of the poses determined in the reference workflow. Spectra of specific voxels or summed spectra of subvolumes were then analysed from these reconstructions in Hyperspy.

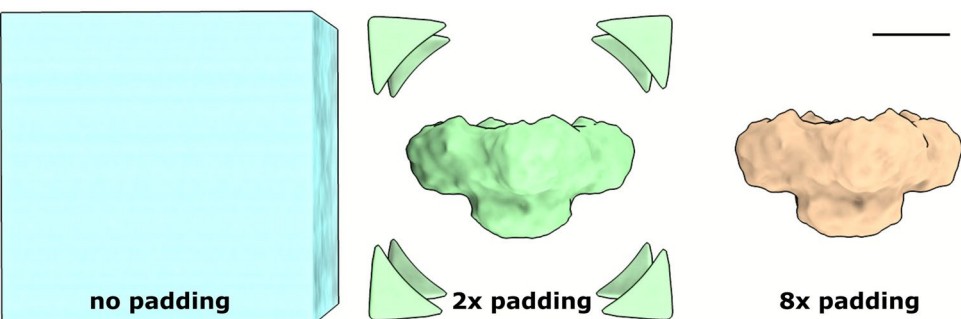

**Extended Data Fig. 5 | Maps reconstructed in a modified version of RELION show different intensities of corner artefacts depending on padding factors.** The reconstruction at 284 eV is shown for three different padding factors (Fourier volume subsampling) with Gaussian filtering with a standard deviation of 7.3 Å applied for better visualisation. No padding yields a strong artefact that dominates the reconstruction. This is reduced by padding. With a padding factor of 2, the artefact is only visible in the corners of the box, while with a padding factor of 8, the artefact does not overlap with the reconstruction at relevant thresholds. The scale bar corresponds to 10 nm and applies to all volumes.

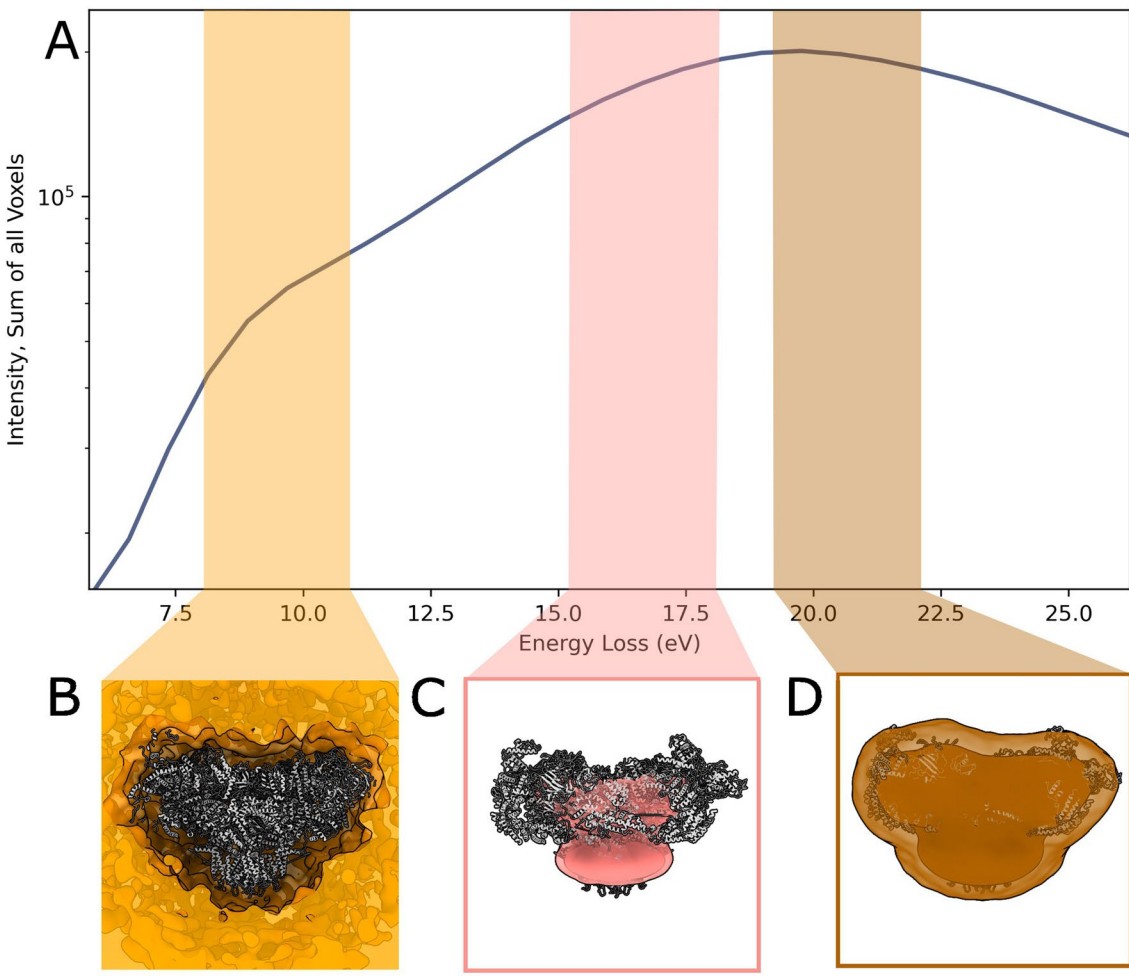

**Extended Data Fig. 6 | Reconstructions from the plasmon region of the energy-loss spectrum show colocalization with solvent, detergent micelle or protein locations at different energy losses.** (**a**) Spectrum of the summed voxels of reconstructions in the plasmon region. (**b-d**) Summed reconstructions for the energy windows (**b**) 8.1 – 11.2 eV, (**c**) 15.1 – 18.2 eV and (**d**) 19.0 – 22.1, each shown at two different visualization thresholds and overlaid on PDB-5TAQ ref. 24, showing different localisations across the volume. The scalebar below panel **b** corresponds to 10 nm and applies to panels **b-d**.

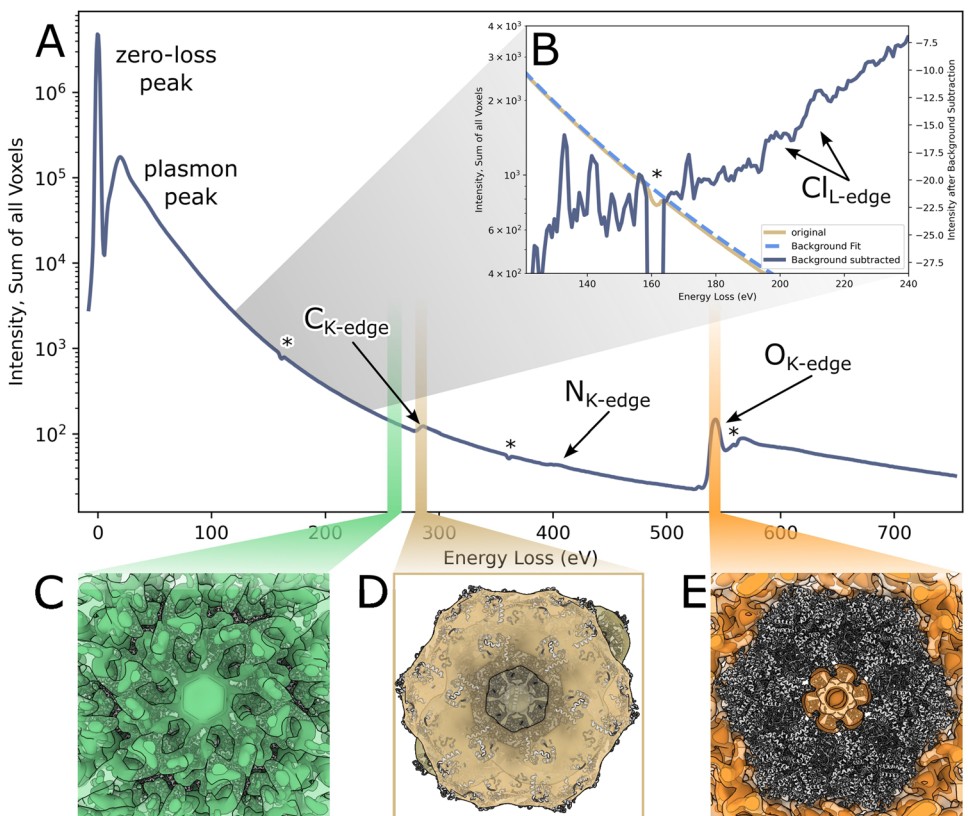

**Extended Data Fig. 7 | Selected spectra and reconstructions of reconstructed EELS data of WH.** (**a**) As for RyR1, the spectrum of the sum of all intensities in the EELS reconstructions shows elemental edges for the most abundant elements: carbon, nitrogen, and oxygen. The asterisks mark regions where the spectra show dips due to double-sized pixels at the tile boundaries of the detector, which are not correctly accounted for by the detector's flatfields. (**b**) For the region between 118 and 240 eV a background subtraction was performed, based on a fit of the spectrum between 121 eV and 127 eV. The subtracted spectrum shows additional edges for phosphorus and chlorine. (**c-e**) Gaussian-filtered reconstructions for three 4.6 eV-regions of the spectrum are shown together with PDB-5M3L ref. 25. Before the carbon edge (**c**), the reconstruction shows uniform distribution of intensity between the protein and the solvent. At the carbon edge (**d**), the reconstruction shows colocalization with the protein, and at the oxygen edge (**e**), the reconstruction shows colocalization with the solvent. The reconstructions reflect the known local distributions of carbon and oxygen. The scalebar near panel **c** corresponds to 10 nm and applies to panels **c-e**.

**Extended Data Table 1 | Cryo-EM data collection, refinement and validation statistics**

|  | Reference RyR1 (EMDB-19191) | Reference WH (EMDB-19190) |
|---|---|---|
| **Data collection and processing** |  |  |
| Magnification | 80.000 | 80.000 |
| Voltage (kV) | 200 | 200 |
| Electron exposure (e–/Å$^2$) | 92 | 92 |
| Defocus range (μm) | 0 | 0 |
| Pixel size (Å) | 3.65 | 3.65 |
| Symmetry imposed | C4 | D6 |
| Initial particle images (no.) | 726609 | 22874 |
| Final particle images (no.) | 460651 | 16648 |
| Map resolution (Å) | 24 | 27 |
| FSC threshold | 0.143 | 0.143 |
| Map resolution range (Å) | 22.4 – 24.7 | 16.7 – 32.3 |

# Reporting Summary

## Statistics

For all statistical analyses, confirm that the following items are present in the figure legend, table legend, main text, or Methods section.

| n/a | Confirmed | |
|---|---|---|
| ☐ | ☒ | The exact sample size (*n*) for each experimental group/condition, given as a discrete number and unit of measurement |
| ☒ | ☐ | A statement on whether measurements were taken from distinct samples or whether the same sample was measured repeatedly |
| ☒ | ☐ | The statistical test(s) used AND whether they are one- or two-sided *Only common tests should be described solely by name; describe more complex techniques in the Methods section.* |
| ☒ | ☐ | A description of all covariates tested |
| ☒ | ☐ | A description of any assumptions or corrections, such as tests of normality and adjustment for multiple comparisons |
| ☒ | ☐ | A full description of the statistical parameters including central tendency (e.g. means) or other basic estimates (e.g. regression coefficient) AND variation (e.g. standard deviation) or associated estimates of uncertainty (e.g. confidence intervals) |
| ☒ | ☐ | For null hypothesis testing, the test statistic (e.g. *F*, *t*, *r*) with confidence intervals, effect sizes, degrees of freedom and *P* value noted *Give P values as exact values whenever suitable.* |
| ☒ | ☐ | For Bayesian analysis, information on the choice of priors and Markov chain Monte Carlo settings |
| ☒ | ☐ | For hierarchical and complex designs, identification of the appropriate level for tests and full reporting of outcomes |
| ☒ | ☐ | Estimates of effect sizes (e.g. Cohen's *d*, Pearson's *r*), indicating how they were calculated |

*Our web collection on statistics for biologists contains articles on many of the points above.*

## Software and code

Policy information about availability of computer code

| Data collection | SerialEM-4.1.0beta5, Panta Rhei version 0.21 |
|---|---|
| Data analysis | Scripts used in the acquisition and processing of the data presented are based upon freely available software packages and are included in the supplementary materials. The following software packages were used for data analysis: crYOLO-1.7.3, RELION-4.0, ChimeraX-1.5, and the Python packages Hyperspy-1.6.4, mrcfile-1.4.0, SciPy-1.7.0, NumPy-1.21.1, and matplotlib-3.4.2. Adobe Premier Pro version 23.6.0 was used for visualisation purposes only. |

For manuscripts utilizing custom algorithms or software that are central to the research but not yet described in published literature, software must be made available to editors and reviewers. We strongly encourage code deposition in a community repository (e.g. GitHub). See the Nature Portfolio guidelines for submitting code & software for further information.

## Data

Policy information about availability of data

All manuscripts must include a data availability statement. This statement should provide the following information, where applicable:

- Accession codes, unique identifiers, or web links for publicly available datasets
- A description of any restrictions on data availability
- For clinical datasets or third party data, please ensure that the statement adheres to our policy

All maps shown in this work have been deposited to publicly accessible databanks. For the RyR1 data set, the reference reconstruction and corresponding half maps, the summed pre-carbon, carbon, nitrogen, and oxygen maps, as well as the background-subtracted carbon and nitrogen maps, and the summed carbon halfmaps were deposited to the EMDB under accession code EMD-19191. The full spectrum of energy-loss reconstructions is available at EDMOND under DOI 10.17617/3.X1R0AQ. Before this is publically released, the data are available for review under the following link: https://edmond.mpg.de/privateurl.xhtml? token=a60b3d8b-825b-436d-a15a-acff7c7b42f0. For the WH data set, the reference reconstruction and corresponding halfmaps, the summed pre-carbon, carbon, and oxygen maps for the WH data set were deposited to the EMDB under accession code EMD-19190. The full spectrum of energy-loss reconstructions is available at EDMOND under DOI 10.17617/3.AUERWM. Prior to public release, the data are available for review under the following link: https://edmond.mpg.de/privateurl.xhtml?token=d4a90bd6-5775-470a-8330-8bbb1bedaf85. In this work, the publicly available structures PDB-5TAQ and PDB-5M3L were used for comparison purposes. The corresponding references are given in the manuscript.

## Research involving human participants, their data, or biological material

Policy information about studies with human participants or human data. See also policy information about sex, gender (identity/presentation), and sexual orientation and race, ethnicity and racism.

| | |
|---|---|
| Reporting on sex and gender | N/A |
| Reporting on race, ethnicity, or other socially relevant groupings | N/A |
| Population characteristics | N/A |
| Recruitment | N/A |
| Ethics oversight | N/A |

Note that full information on the approval of the study protocol must also be provided in the manuscript.

# Field-specific reporting

Please select the one below that is the best fit for your research. If you are not sure, read the appropriate sections before making your selection.

☒ Life sciences  ☐ Behavioural & social sciences  ☐ Ecological, evolutionary & environmental sciences

For a reference copy of the document with all sections, see nature.com/documents/nr-reporting-summary-flat.pdf

# Life sciences study design

All studies must disclose on these points even when the disclosure is negative.

| | |
|---|---|
| Sample size | The dataset size was determined by available instrument time. |
| Data exclusions | Particle images were excluded on the basis of visual inspection of 2D class averages, as established in the field. |
| Replication | Data were collected and analyzed for two different protein complexes, rabbit muscle ryanodine receptor (RyR1) and worm hemoglobin. Visual analysis of single-voxel spectra was carried out for spectra of at least twenty randomly chosen voxels in the protein area and at least twenty randomly chosen voxels outside the protein area. 1142 spectral images of RyR1 and 61 spectral images of worm hemoglobin were collected. |
| Randomization | For 3D reconstruction, particle images were automatically assigned to random halves of the dataset by the reconstruction software. This random assignment was maintained for all corresponding energy loss reconstructions. |
| Blinding | For the comparison of annular dark field, elastic bright field and combined modes of image formation for reference reconstruction, no blinding was carried out because the assessment of these groups relied on an objective criterion, the map-to-model FSC. No blinding was possible for selection of particle images on the basis of 2D class averages, as experimenter judgement is required for this step, as established in the field. No other comparisons between different experimental groups or between experimental and control groups are described, therefore no blinding was required for other aspects of the study. |

# Behavioural & social sciences study design

All studies must disclose on these points even when the disclosure is negative.

| | |
|---|---|
| Study description | N/A |
| Research sample | N/A |
| Sampling strategy | N/A |
| Data collection | N/A |
| Timing | N/A |
| Data exclusions | N/A |
| Non-participation | N/A |
| Randomization | N/A |

# Ecological, evolutionary & environmental sciences study design

All studies must disclose on these points even when the disclosure is negative.

| | |
|---|---|
| Study description | N/A |
| Research sample | N/A |
| Sampling strategy | N/A |
| Data collection | N/A |
| Timing and spatial scale | N/A |
| Data exclusions | N/A |
| Reproducibility | N/A |
| Randomization | N/A |
| Blinding | N/A |

Did the study involve field work?  ☐ Yes  ☐ No

## Field work, collection and transport

| | |
|---|---|
| Field conditions | N/A |
| Location | N/A |
| Access & import/export | N/A |
| Disturbance | N/A |

# Reporting for specific materials, systems and methods

We require information from authors about some types of materials, experimental systems and methods used in many studies. Here, indicate whether each material, system or method listed is relevant to your study. If you are not sure if a list item applies to your research, read the appropriate section before selecting a response.

## Materials & experimental systems

| n/a | Involved in the study |
|-----|------------------------|
| ☒ ☐ | Antibodies |
| ☒ ☐ | Eukaryotic cell lines |
| ☒ ☐ | Palaeontology and archaeology |
| ☐ ☒ | Animals and other organisms |
| ☒ ☐ | Clinical data |
| ☒ ☐ | Dual use research of concern |
| ☒ ☐ | Plants |

## Methods

| n/a | Involved in the study |
|-----|------------------------|
| ☒ ☐ | ChIP-seq |
| ☒ ☐ | Flow cytometry |
| ☒ ☐ | MRI-based neuroimaging |

# Antibodies

| | |
|---|---|
| Antibodies used | *Describe all antibodies used in the study; as applicable, provide supplier name, catalog number, clone name, and lot number.* |
| Validation | *Describe the validation of each primary antibody for the species and application, noting any validation statements on the manufacturer's website, relevant citations, antibody profiles in online databases, or data provided in the manuscript.* |

# Eukaryotic cell lines

Policy information about cell lines and Sex and Gender in Research

| | |
|---|---|
| Cell line source(s) | *State the source of each cell line used and the sex of all primary cell lines and cells derived from human participants or vertebrate models.* |
| Authentication | *Describe the authentication procedures for each cell line used OR declare that none of the cell lines used were authenticated.* |
| Mycoplasma contamination | *Confirm that all cell lines tested negative for mycoplasma contamination OR describe the results of the testing for mycoplasma contamination OR declare that the cell lines were not tested for mycoplasma contamination.* |
| Commonly misidentified lines (See ICLAC register) | *Name any commonly misidentified cell lines used in the study and provide a rationale for their use.* |

# Palaeontology and Archaeology

| | |
|---|---|
| Specimen provenance | *Provide provenance information for specimens and describe permits that were obtained for the work (including the name of the issuing authority, the date of issue, and any identifying information). Permits should encompass collection and, where applicable, export.* |
| Specimen deposition | *Indicate where the specimens have been deposited to permit free access by other researchers.* |
| Dating methods | *If new dates are provided, describe how they were obtained (e.g. collection, storage, sample pretreatment and measurement), where they were obtained (i.e. lab name), the calibration program and the protocol for quality assurance OR state that no new dates are provided.* |

☐ Tick this box to confirm that the raw and calibrated dates are available in the paper or in Supplementary Information.

| | |
|---|---|
| Ethics oversight | *Identify the organization(s) that approved or provided guidance on the study protocol, OR state that no ethical approval or guidance was required and explain why not.* |

Note that full information on the approval of the study protocol must also be provided in the manuscript.

# Animals and other research organisms

Policy information about studies involving animals; ARRIVE guidelines recommended for reporting animal research, and Sex and Gender in Research

| | |
|---|---|
| Laboratory animals | Rabbit tissue was sourced from a commercial supplier. Earthworms were sourced from a commercial supplier. The age of the (mature) earthworms was unknown and is not relevant to the study. |
| Wild animals | No wild animals were used in this study. |
| Reporting on sex | *Indicate if findings apply to only one sex; describe whether sex was considered in study design, methods used for assigning sex. Provide data disaggregated for sex where this information has been collected in the source data as appropriate; provide overall numbers in this Reporting Summary. Please state if this information has not been collected. Report sex-based analyses where performed, justify reasons for lack of sex-based analysis.* |

| | |
|---|---|
| Field-collected samples | No field-collected samples were used in this study. |
| Ethics oversight | This study did not require ethical approval. |

Note that full information on the approval of the study protocol must also be provided in the manuscript.

# Clinical data

Policy information about clinical studies

All manuscripts should comply with the ICMJE guidelines for publication of clinical research and a completed CONSORT checklist must be included with all submissions.

| | |
|---|---|
| Clinical trial registration | *Provide the trial registration number from ClinicalTrials.gov or an equivalent agency.* |
| Study protocol | *Note where the full trial protocol can be accessed OR if not available, explain why.* |
| Data collection | *Describe the settings and locales of data collection, noting the time periods of recruitment and data collection.* |
| Outcomes | *Describe how you pre-defined primary and secondary outcome measures and how you assessed these measures.* |

# Dual use research of concern

Policy information about dual use research of concern

## Hazards

Could the accidental, deliberate or reckless misuse of agents or technologies generated in the work, or the application of information presented in the manuscript, pose a threat to:

No | Yes
☐ | ☐ Public health
☐ | ☐ National security
☐ | ☐ Crops and/or livestock
☐ | ☐ Ecosystems
☐ | ☐ Any other significant area

## Experiments of concern

Does the work involve any of these experiments of concern:

No | Yes
☐ | ☐ Demonstrate how to render a vaccine ineffective
☐ | ☐ Confer resistance to therapeutically useful antibiotics or antiviral agents
☐ | ☐ Enhance the virulence of a pathogen or render a nonpathogen virulent
☐ | ☐ Increase transmissibility of a pathogen
☐ | ☐ Alter the host range of a pathogen
☐ | ☐ Enable evasion of diagnostic/detection modalities
☐ | ☐ Enable the weaponization of a biological agent or toxin
☐ | ☐ Any other potentially harmful combination of experiments and agents

# Plants

| | |
|---|---|
| Seed stocks | *Report on the source of all seed stocks or other plant material used. If applicable, state the seed stock centre and catalogue number. If plant specimens were collected from the field, describe the collection location, date and sampling procedures.* |
| Novel plant genotypes | *Describe the methods by which all novel plant genotypes were produced. This includes those generated by transgenic approaches, gene editing, chemical/radiation-based mutagenesis and hybridization. For transgenic lines, describe the transformation method, the number of independent lines analyzed and the generation upon which experiments were performed. For gene-edited lines, describe the editor used, the endogenous sequence targeted for editing, the targeting guide RNA sequence (if applicable) and how the editor was applied.* |
| Authentication | *Describe any authentication procedures for each seed stock used or novel genotype generated. Describe any experiments used to assess the effect of a mutation and, where applicable, how potential secondary effects (e.g. second site T-DNA insertions, mosiacism, off-target gene editing) were examined.* |

# ChIP-seq

## Data deposition

☐ Confirm that both raw and final processed data have been deposited in a public database such as GEO.

☐ Confirm that you have deposited or provided access to graph files (e.g. BED files) for the called peaks.

| | |
|---|---|
| **Data access links** <br> *May remain private before publication.* | *For "Initial submission" or "Revised version" documents, provide reviewer access links. For your "Final submission" document, provide a link to the deposited data.* |
| **Files in database submission** | *Provide a list of all files available in the database submission.* |
| **Genome browser session** <br> (e.g. UCSC) | *Provide a link to an anonymized genome browser session for "Initial submission" and "Revised version" documents only, to enable peer review. Write "no longer applicable" for "Final submission" documents.* |

## Methodology

| | |
|---|---|
| **Replicates** | *Describe the experimental replicates, specifying number, type and replicate agreement.* |
| **Sequencing depth** | *Describe the sequencing depth for each experiment, providing the total number of reads, uniquely mapped reads, length of reads and whether they were paired- or single-end.* |
| **Antibodies** | *Describe the antibodies used for the ChIP-seq experiments; as applicable, provide supplier name, catalog number, clone name, and lot number.* |
| **Peak calling parameters** | *Specify the command line program and parameters used for read mapping and peak calling, including the ChIP, control and index files used.* |
| **Data quality** | *Describe the methods used to ensure data quality in full detail, including how many peaks are at FDR 5% and above 5-fold enrichment.* |
| **Software** | *Describe the software used to collect and analyze the ChIP-seq data. For custom code that has been deposited into a community repository, provide accession details.* |

# Flow Cytometry

## Plots

Confirm that:

☐ The axis labels state the marker and fluorochrome used (e.g. CD4-FITC).

☐ The axis scales are clearly visible. Include numbers along axes only for bottom left plot of group (a 'group' is an analysis of identical markers).

☐ All plots are contour plots with outliers or pseudocolor plots.

☐ A numerical value for number of cells or percentage (with statistics) is provided.

## Methodology

| | |
|---|---|
| **Sample preparation** | *Describe the sample preparation, detailing the biological source of the cells and any tissue processing steps used.* |
| **Instrument** | *Identify the instrument used for data collection, specifying make and model number.* |
| **Software** | *Describe the software used to collect and analyze the flow cytometry data. For custom code that has been deposited into a community repository, provide accession details.* |
| **Cell population abundance** | *Describe the abundance of the relevant cell populations within post-sort fractions, providing details on the purity of the samples and how it was determined.* |
| **Gating strategy** | *Describe the gating strategy used for all relevant experiments, specifying the preliminary FSC/SSC gates of the starting cell population, indicating where boundaries between "positive" and "negative" staining cell populations are defined.* |

☐ Tick this box to confirm that a figure exemplifying the gating strategy is provided in the Supplementary Information.

# Magnetic resonance imaging

## Experimental design

| | |
|---|---|
| **Design type** | *Indicate task or resting state; event-related or block design.* |

| Design specifications | *Specify the number of blocks, trials or experimental units per session and/or subject, and specify the length of each trial or block (if trials are blocked) and interval between trials.* |
|---|---|
| Behavioral performance measures | *State number and/or type of variables recorded (e.g. correct button press, response time) and what statistics were used to establish that the subjects were performing the task as expected (e.g. mean, range, and/or standard deviation across subjects).* |

## Acquisition

| Imaging type(s) | *Specify: functional, structural, diffusion, perfusion.* |
|---|---|
| Field strength | *Specify in Tesla* |
| Sequence & imaging parameters | *Specify the pulse sequence type (gradient echo, spin echo, etc.), imaging type (EPI, spiral, etc.), field of view, matrix size, slice thickness, orientation and TE/TR/flip angle.* |
| Area of acquisition | *State whether a whole brain scan was used OR define the area of acquisition, describing how the region was determined.* |

Diffusion MRI    ☐ Used    ☐ Not used

## Preprocessing

| Preprocessing software | *Provide detail on software version and revision number and on specific parameters (model/functions, brain extraction, segmentation, smoothing kernel size, etc.).* |
|---|---|
| Normalization | *If data were normalized/standardized, describe the approach(es): specify linear or non-linear and define image types used for transformation OR indicate that data were not normalized and explain rationale for lack of normalization.* |
| Normalization template | *Describe the template used for normalization/transformation, specifying subject space or group standardized space (e.g. original Talairach, MNI305, ICBM152) OR indicate that the data were not normalized.* |
| Noise and artifact removal | *Describe your procedure(s) for artifact and structured noise removal, specifying motion parameters, tissue signals and physiological signals (heart rate, respiration).* |
| Volume censoring | *Define your software and/or method and criteria for volume censoring, and state the extent of such censoring.* |

## Statistical modeling & inference

| Model type and settings | *Specify type (mass univariate, multivariate, RSA, predictive, etc.) and describe essential details of the model at the first and second levels (e.g. fixed, random or mixed effects; drift or auto-correlation).* |
|---|---|
| Effect(s) tested | *Define precise effect in terms of the task or stimulus conditions instead of psychological concepts and indicate whether ANOVA or factorial designs were used.* |

Specify type of analysis:    ☐ Whole brain    ☐ ROI-based    ☐ Both

| Statistic type for inference<br><br>(See Eklund et al. 2016) | *Specify voxel-wise or cluster-wise and report all relevant parameters for cluster-wise methods.* |
|---|---|
| Correction | *Describe the type of correction and how it is obtained for multiple comparisons (e.g. FWE, FDR, permutation or Monte Carlo).* |

## Models & analysis

n/a | Involved in the study
☐ | ☐ Functional and/or effective connectivity
☐ | ☐ Graph analysis
☐ | ☐ Multivariate modeling or predictive analysis

| Functional and/or effective connectivity | *Report the measures of dependence used and the model details (e.g. Pearson correlation, partial correlation, mutual information).* |
|---|---|
| Graph analysis | *Report the dependent variable and connectivity measure, specifying weighted graph or binarized graph, subject- or group-level, and the global and/or node summaries used (e.g. clustering coefficient, efficiency, etc.).* |
| Multivariate modeling and predictive analysis | *Specify independent variables, features extraction and dimension reduction, model, training and evaluation metrics.* |

