## [Peer Review File · Nature Methods]

Elemental mapping in single-particle reconstructions by Reconstructed Electron Energy-Loss (REEL) analysis

Corresponding Author: Dr Bonnie Murphy

Version 0:

Decision Letter:

12th Mar 2024

Dear Bonnie,

Your Article, "Localising elements in single-particle reconstructions by REEL-EM: Reconstructed Electron Energy-Loss - Elemental Mapping", has now been seen by 4 reviewers. As you will see from their comments below, although the reviewers find your work of considerable potential interest, they have raised some concerns. We are interested in the possibility of publishing your paper in Nature Methods, but would like to consider your response to these concerns before we reach a final decision on publication.

We therefore invite you to revise your manuscript to address these concerns. We think most of them can be addressed by making clarifications or adding discussion to the text. It is optional for you to include additional data on other protein complexes, but we do ask you to discuss broader applicability and limitations of the method, including to smaller complexes.

Link Redacted

We hope to receive your revised paper within 6 weeks. If you cannot send it within this time, please let us know. In this event, we will still be happy to reconsider your paper at a later date so long as nothing similar has been accepted for publication at Nature Methods or published elsewhere.

OPEN SCIENCE REQUIREMENTS

REPORTING SUMMARY AND EDITORIAL POLICY CHECKLISTS

IMAGE INTEGRITY

DATA AVAILABILITY

All novel DNA and RNA sequencing data, protein sequences, genetic polymorphisms, linked genotype and phenotype data, gene expression data, macromolecular structures, and proteomics data must be deposited in a publicly accessible database, and accession codes and associated hyperlinks must be provided in the "Data Availability" section.

CODE AVAILABILITY

Please include a "Code Availability" subsection in the Online Methods which details how your custom code is made available. Only in rare cases (where code is not central to the main conclusions of the paper) is the statement "available upon request" allowed (and reasons should be specified).

MATERIALS AVAILABILITY

SUPPLEMENTARY PROTOCOL

To help facilitate reproducibility and uptake of your method, we ask you to prepare a step-by-step Supplementary Protocol for the method described in this paper. We [encourage authors to share their step-by-step experimental protocols](https://www.nature.com/nature-research/editorial-policies/reporting-standards#protocols) on a protocol sharing platform of their choice and report the protocol DOI in the reference list. Nature Portfolio's Protocol Exchange is a free-to-use and open resource for protocols; protocols deposited in Protocol Exchange are citable and can be linked from the published article. More details can be found at www.nature.com/protocolexchange/about.

ORCID

Sincerely yours,
Allison

Allison Doerr, Ph.D.
Chief Editor
Nature Methods

Reviewers' Comments:

Reviewer #1:

Remarks to the Author:

A. This study by O. Pfeil-Gardiner et al. on "Localizing elements in single-particle reconstructions by REEL-EM: Reconstructed electron energy loss elemental mapping presents data from a state-of-the-art Titan Krios G3 transmission electron microscope operated in the scanning transmission EM (STEM) mode and equipped with a CEOS-CEFID energy filter and Dectris ELA hybrid pixel detector. The authors set out to image two biomolecular complexes (the ryanodine receptor RyR1 and earthworm hemoglobin) at a low electron dose of approximately 100 electrons per square Angstrom, using a combination of single particle analysis (SPA) in cryo-EM and electron spectroscopic imaging.

The authors demonstrate that the zero-loss spectral peak provides a usable elastic bright field signal from which it is feasible to determine single-particle analyses of the protein assemblies, while simultaneously acquiring the weak core loss signals from specific atoms in the structure such as carbon, nitrogen, oxygen, and phosphorus. The study does demonstrate that feasibility of low-dose elemental mapping, the spatial resolution is only about 20-40 Angstroms.

B. The idea of performing EELS mapping of specific elements in biomolecular complexes has been discussed previously, but the current paper describes the first report of a realistic approach for achieving this goal although, as the authors indicate, single

atom sensitivity and resolution are not yet realized in this study.

C. The data, methodology and approach used by the authors is of high quality and represents the state-of-the-art in terms of instrumentations and computational analysis.

D. Statistical treatment of the data is appropriate for the study.

E and F. Conclusions and data analysis methods might be supplement based on comments below.

G. References are appropriate.

H. Clarity and context are good for the most part, although some additions could be considered as indicated below.

Major comment:

It would be helpful if the authors could compute a thickness image of the specimen expressed in numbers of inelastic mean free paths, i.e., thickness/mean free path = $\ln(I_{\text{total}}/I_0)$, where I_{total} = total intensity in the energy loss spectrum including the zero-loss peak, and I_0 = the zero-loss intensity. This would provide a measure of whether plural inelastic scattering is changing from pixel to pixel in the image. If the thickness map is almost constant, then it might be possible to compute the elemental maps with much less noise, based on the assumption of a constant shape for the background underlying a core edge. In that case, it would be feasible to compute the mean "r-value", i.e., the constant of the inverse power law of the background as a function of energy loss E underlying a particular core edge ($=AE^{-r}$). Then it might be possible to subtract a constant x the pre-edge background intensity from the post-edge intensity, instead of trying to fit a power law that is unconstrained, and very noisy. This is explained in an early STEM-EELS paper (Leapman R. Scanning Transmission Electron Microscope (STEM) Elemental Mapping by Electron Energy-Loss Spectroscopy, 1986; Annals of the New York Academy of Sciences 483 (1), 326-338). Although this reference shows that artifacts can be produced from the false assumption that the background shape is constant, there are situations where the assumption could be tested to see if it holds. This reviewer suspects that the present study could be one of those situations. Also, if small numbers of atoms become visible reproducibly, then it's unlikely that the amount of plural scattering could produce a plural inelastic scattering artifact.

Other comments:

(1) On page 2, the authors should make it clear that the 3D information about the biomolecular complexes is contained in 2D images of large numbers of identical structures in different orientations in the SPA approach. Some readers who are not familiar with SPA might imagine that a tomographic approach is being taken.

(2) Typically, the electron dose for SPA analysis is 10-30 electrons/square angstrom. Could the authors comment on the effect of increasing the dose to 100 electrons/square angstrom, for detecting single atoms of metals and other elements in REEL-EM? The resolution achieved in the present study of 20-40 angstroms for the RyR1 and WH structures seems to be quite modest.

(3) The authors should include scale bars in all their RyR1 and WH model structures (Fig. 2, Fig. 3, Fig. 5 C,D,E, and Fig. 6.

(4) This reviewer might have missed seeing it, but the authors should ensure that they include the molecular weights of the RyR1 and WH complexes.

(5) It would be helpful if the authors could comment further on the relative advantages of the ADF and EBF images? The authors suggest that neither of these signals provides the contrast seen in standard phase contrast SPA. To what extent can high quality SPA analysis be combined with REEL analysis?

Reviewer #2:

Remarks to the Author:

Pfeil-Gardiner et al. describe the implementation of electron energy loss spectroscopy with a focused electron beam, scanning over a frozen hydrated cryo-EM specimen in STEM mode. At each pixel position of the electron beam probe, an entire electron energy loss spectrum is recorded with the help of a CEOS energy filter, behind which a Dectris ELA hybrid pixel detector is used to record the spectra. At least initially, a dark-field ring detector was also used to quantify the elastically scattered electrons at higher scattering angles, which give access to a zero-loss HAADF STEM image. However, the authors found that this was very noisy, and a more useful image could be generated by producing a STEM image from the zero-loss electrons in the EELS spectrum. This is then an image, where the absence of electrons indicates the presence of a protein, because at these locations the protein had scattered the electrons out of the EELS spectrum and to higher angles on the HAADF ring detector.

The zero-loss EELS image is first used to reconstruct the structure of the proteins, here using as test specimen the (huge) ryanodine receptor, and also haemoglobin particles. With the help of the particle metadata from that reconstruction, such as particle positions and particle orientations, additional 3D reconstructions for the particles were then generated, using images generated only from electrons that had undergone a specific energy loss. For this, the EELS spectra were sampled in energy

bins of 0.77 eV, resulting in sets of images for the spectra that cover -25.4 eV to +770 eV energy loss. These many sets of energy loss images were then used to generate a series of 3D protein reconstructions, each created from the signal within a specific electron energy loss window. Since certain elements in the sample, such as carbon, potassium, nitrogen or oxygen, etc., and also higher weight elements such as phosphorus or chloride cause specific energy losses to passing electrons, the images generated from electrons that suffered those characteristic energy losses can then be used to localize those specific atoms in the protein structure. In conclusion, this STEM-EELS method for the first time results in a cryo-EM tool that can localize specific types of atoms in the 3D structure of the single particle proteins.

The obtained resolutions for the two test samples are very modest. The zero-loss EELS images must have a very low signal-to-noise ratio to result in a 24 Å 3D reconstruction from almost half a million particles. One such image is shown in Figure 3B, which illustrates the low SNR of the method. The authors should specify the molecular weight of the imaged particles. The RyR1 is over 3 MDa, if I remember well. What is the weight of the hemoglobin particles? How is the likelihood that this STEM EELS method can also be applied to smaller particles, such as typical 100kDa particles that are otherwise investigated a lot in structural biology? This should be discussed in this manuscript.

Ferritin particles filled with an iron core might be an interesting test object for follow-up works (in another paper), or bacteria that accumulate heavy metals, such as uranium. This would then need combination with tomography, but the thicker sample might be well suitable for EELS STEM approaches. Such studies should remain for follow-up works, but could be mentioned here as outlook in the discussion.

A higher resolution might be reachable, if the STEM EELS method would be combined with a phase contrast method, such as defocused CTEM or iDPC imaging. The authors themselves discuss this correctly on Page 8 in lines 10-15. This reviewer, nevertheless, agrees with the authors that this present manuscript is a first proof of concept, which is a milestone achievement by itself and worth its own publication. Combination of the new EELS single particle concept with more routine methods, such as conventional defocused CTEM or the more recent iDPC or ptychography method, is an interesting task for a follow-up projects.

The manuscript is excellently written. It is well structured, interesting to read, has clear figures, and no significant redundancy.

Some details:

Page 2, Introduction: The first paragraph repeats "even though" twice in the same sentence. Please rephrase.

Page 2, line 33 (approximately, line numbers were unfortunately missing): You use e^-/A^2 for the first time here. Please define, what that unit refers to. Please also define here, what a normally allowed dose budget would correspond to. You define that later ($100 e^-/A^2$), but it would be more useful to have already here in the text.

Page 3: line 1: You call this method REEL-EM, for "Reconstructed Electron Energy-Loss Elemental Mapping". I find this is an unfortunate abbreviation, because in the context of this story, EM usually stands for electron microscopy. You define cryo-EM in the first line of the introduction. Now you used REEL-EM here with EM standing for Elemental Mapping. I suggest finding a different way to abbreviate this, or use a different name for the method.

Henning Stahlberg.

Reviewer #3:

Remarks to the Author:

A. Summary of key results

The article presents a method for reconstructing 3D spatially resolved EELS data, and thus elemental information, obtained from biological samples.

EELS is an established technique for elemental analysis, however, it requires high doses to achieve sufficient signal and is hence difficult to apply to radiation-sensitive biological samples. The presented technique combines EELS with a single-particle analysis cryoEM workflow, which yields accurate particle coordinates and poses that can be used to extract the energy-loss information from the spectral data. This step distributes the total dose over a large number of particles and thus allows simultaneous spatial and spectral reconstruction from the elastically (zero loss peak) and inelastically scattered electrons, respectively, resulting in a reconstructed 3D volume for each energy bin.

The method is applied to two proteins in a proof-of-principle study for which such 4D datasets including both spatial and spectral information could be obtained. The distribution of abundant elements, such as carbon, could be reconstructed at a resolution sufficient for comparison with higher-resolution structures obtained from conventional cryoEM. The carbon edge could also be discerned in per-voxel spectra, pointing to the possibility of single-atom sensitivity (1-2 C atoms per voxel). Full single-atom sensitivity of non-abundant species could so far not be shown.

B. Originality and Significance

Originality

While previous studies have applied EELS and related techniques to biological specimens, the approach combining EELS and SPA is novel.

Significance

The method addresses an important problem in cryoEM, namely the lack of chemical specificity, which can hinder the accurate assignment of densities especially regarding the localisation of metal ions and small ligands. Hence a technique introducing elemental sensitivity into cryoEM would be highly relevant for many biological applications. While the presented data does not yet address a specific biological question, it serves as a proof-of-principle, thereby demonstrating the potential of the technique. One point to make here is that the elements mapped in this paper (mainly C, N and O in the solvent), are so abundant in biological specimens, that they will probably not lead to many major insights. This being said, everything has to start somewhere and this is an interesting application.

C. Data and methodology

As this is a proof-of-principle paper, the presented data validates the approach taken. The cryoEM results are validated in the standard ways. The quality of the data set is sufficient to demonstrate that the method works on abundant elements. The ways in which improved data quality could contribute to achieving the goal of single-atom sensitivity for non-abundant elements is discussed.

The presentation of the method and the results is clear.

D. Appropriate use of statistics and treatment of uncertainties

Not applicable?

E. Conclusions: robustness, validity, reliability

The conclusions drawn appear robust, valid and reliable. The current scope of the method is clearly presented and not overstated.

F. Suggested improvements: experiments, data for possible revision

- 1) To give the reader a better impression of the information contained in the spectral images and how it relates to the information from the bright-field images, the authors should add a larger scale example of a spectral image than the one shown in Fig. 1.
- 2) Supplementary movies: While the changes observed at both the carbon and oxygen edges are very clear, the interpretation of other parts of the spectrum and the corresponding reconstructed volume should be clarified:
 - Why do the molecules not appear in the zero-loss peak? Is it because of the DF imaging?
 - The plasmon peak region is interesting, can the different localisations shown there (also Fig S5) be interpreted in a biologically relevant way?
 - In movie 2, both the region between plasmon peak and C edge as well as the trailing C edge itself show interesting geometrical changes, can this be interpreted?
 - In Figure 6, the N edge is discernible, although not as strongly as the C edge, in the movie, it is not discernible. Is this due to the remaining carbon signal? If so, can an estimate be made how strong the signal would need to be to be discernible in the movie? A similar question pertains to the P and Cl edges shown in Figure 5.
- 3) On page 6, it is stated that "Of the known elements in our sample we detect all except hydrogen, zinc, sulphur, and calcium." For clarification, it should be stated which elements are present in the sample and/or which elements have been detected.
- 4) In the same paragraph, parts of Figure 5 seem to have been misreferenced (5B and 5C in the text, whereas the discussion appears to refer to panels 5F and 5G).
- 5) Discussion of dose at end of the same paragraph (page 6, top): It would be beneficial to add some discussion here (some of it is addressed in the Discussion section) regarding how these inaccuracies in the poses could be improved, e.g. the role of higher accumulated dose (larger datasets), would lower dose per particle help with pose accuracy? See also 9).
- 6) Single-voxel spectra (Fig 5, Fig 6): The presented voxels in Fig 5 and 6 appear to be the same. If so, this should be stated. How many voxels have been studied in this way? Are they all featuring similar characteristics? Could conclusions be drawn from comparisons, e.g. between a voxel in the protein bulk and the micelle region, about abundance of nitrogen, and hence, by extrapolation, about the number of atoms of a species in a given set of neighbouring voxels to get a reliable edge signal?
- 7) In Figure S6, the grey area on the spectrum extends further than the range shown in the inset.
- 8) The Discussion section is largely a summary plus an outlook regarding the some of the current problems and the future development of the method, given that, the authors may want to consider renaming this section. As already suggested in point 5), the authors may further want to consider moving the discussion regarding total dose and pose inaccuracies to the relevant place in Section Reconstruction of energy-loss data generates 3D EELS maps (page 6) and focus on future development in the section currently named Discussion.
- 9) Regarding the future development of the technique, the authors list many different approaches. It would be beneficial to clarify which of these the authors think are the most likely to significantly improve the method in the near future.
- 10) Choice of test samples: were there particular reasons to choose these proteins as test samples (other than the presence of metal ligands)? Would it make sense to test the method on standard cryoEM samples? To test sensitivity, could proteins with more than one metal ligand at a binding site be used?
- 11) References 34 and 35 appear to be the same.

G. References: appropriate credit to previous work?

Largely, the reference list gives a good picture of past work. However, the authors could perhaps cite Herdman et al Structure 2022 where single-particle EM was used to map single Holmium/Calcium ions. Another article to perhaps cite is Wu et al Science 2012, where beam damage was used to distinguish between protein and DNA.

H. Clarity and context: lucidity of abstract/summary, appropriateness of abstract, introduction and conclusions

Abstract, Introduction and conclusions are all appropriate and clear. As the section titled Discussion is mostly contains a summary, conclusions and an outlook, the authors may want to consider renaming this section as suggested above.

Reviewer #4:

None

Editor's note: Reviewer #4 co-reviewed with Reviewer #3 and therefore did not provide separate comments to the authors.

Version 1:

Decision Letter:

Our ref: NMETH-A54782A

4th Jun 2024

Dear Bonnie,

Thank you for submitting your revised manuscript "Elemental mapping in single-particle reconstructions by Reconstructed Electron Energy-Loss (REEL) analysis" (NMETH-A54782A). It has now been seen by the original referees and their comments are below. The reviewers find that the paper has improved in revision, and therefore we'll be happy in principle to publish it in Nature Methods, pending minor revisions to satisfy the referees' final requests and to comply with our editorial and formatting guidelines.

TRANSPARENT PEER REVIEW

ORCID

Sincerely yours,
Allison

Allison Doerr, Ph.D.
Chief Editor
Nature Methods

Reviewer #1 (Remarks to the Author):

The authors have satisfactorily addressed the questions raised by this reviewer.

Although the current paper describes a proof-of-principle technique, it could become a valuable approach and so is recommended for publication.

Reviewer #2 (Remarks to the Author):

The authors have addressed all my earlier comments to my fullest satisfaction.

Reviewer #3 (Remarks to the Author):

I recommend publication of the manuscript.

Reviewer #4 (Remarks to the Author):

The authors have addressed all points raised by the reviewers well and have made appropriate changes. The present version of the manuscript is suitable for publication.

As a very minor point, the authors may want to consider mentioning the statistics on the consistency of the single-voxel spectra across the protein reported in their response to Reviewer #3, point 6 either in the main text, figure caption or methods section as this would strengthen the point they are making.

Version 2:

Decision Letter:

23rd Sep 2024

Dear Bonnie,

I am pleased to inform you that your Article, "Elemental mapping in single-particle reconstructions by Reconstructed Electron Energy-Loss (REEL) analysis", has now been accepted for publication in *Nature Methods*. The received and accepted dates will be 16 January 2024 and 23 September 2024. This note is intended to let you know what to expect from us over the next month or so, and to let you know where to address any further questions.

Over the next few weeks, your paper will be copyedited to ensure that it conforms to *Nature Methods* style. Once your paper is typeset, you will receive an email with a link to choose the appropriate publishing options for your paper and our Author Services team will be in touch regarding any additional information that may be required. It is extremely important that you let us know now whether you will be difficult to contact over the next month. If this is the case, we ask that you send us the contact information (email, phone and fax) of someone who will be able to check the proofs and deal with any last-minute problems.

Please note that *Nature Methods* is a Transformative Journal (TJ). Authors may publish their research with us through the traditional subscription access route or make their paper immediately open access through payment of an article-processing charge (APC). Authors will not be required to make a final decision about access to their article until it has been accepted. [Find out more about Transformative Journals](https://www.springernature.com/gp/open-research/transformative-journals)

If you are active on Twitter/X, please e-mail me your and your coauthors' handles so that we may tag you when the paper is published.

Best regards,
Allison

Allison Doerr, Ph.D.
Chief Editor
Nature Methods

** Visit the Springer Nature Editorial and Publishing website at http://editorial-jobs.springernature.com?utm_source=ejP_NMeth_email&utm_medium=ejP_NMeth_email&utm_campaign=ejp_Nmeth for more information about our career opportunities. If you have any questions please click [here](mailto:editorial.publishing.jobs@springernature.com).**

permits use, sharing, adaptation, distribution and reproduction in any medium or format, as long as you give appropriate credit to the original author(s) and the source, provide a link to the Creative Commons license, and indicate if changes were made. In cases where reviewers are anonymous, credit should be given to 'Anonymous Referee' and the source.

Response to reviewers' comments

We thank the reviewers for their positive assessment of our manuscript and for their thoughts on areas of potential improvement. We have used the feedback to improve our manuscript, and include below a point-by-point summary of changes.

Reviewers' Comments:

Reviewer #1:

Remarks to the Author:

A. This study by O. Pfeil-Gardiner et al. on "Localizing elements in single-particle reconstructions by REEL-EM: Reconstructed electron energy loss elemental mapping presents data from a state-of-the-art Titan Krios G3 transmission electron microscope operated in the scanning transmission EM (STEM) mode and equipped with a CEOS-CEFID energy filter and Dectris ELA hybrid pixel detector. The authors set out to image two biomolecular complexes (the ryanodine receptor RyR1 and earthworm hemoglobin) at a low electron dose of approximately 100 electrons per square Angstrom, using a combination of single particle analysis (SPA) in cryo-EM and electron spectroscopic imaging.

The authors demonstrate that the zero-loss spectral peak provides a usable elastic bright field signal from which it is feasible to determine single-particle analyses of the protein assemblies, while simultaneously acquiring the weak core loss signals from specific atoms in the structure such as carbon, nitrogen, oxygen, and phosphorus. The study does demonstrate that feasibility of low-dose elemental mapping, the spatial resolution is only about 20-40 Angstroms.

B. The idea of performing EELS mapping of specific elements in biomolecular complexes has been discussed previously, but the current paper describes the first report of a realistic approach for achieving this goal although, as the authors indicate, single atom sensitivity and resolution are not yet realized in this study.

C. The data, methodology and approach used by the authors is of high quality and represents the state-of-the-art in terms of instrumentations and computational analysis.

D. Statistical treatment of the data is appropriate for the study.

E and F. Conclusions and data analysis methods might be supplemented based on comments below.

G. References are appropriate.

H. Clarity and context are good for the most part, although some additions could be considered as indicated below.

Major comment:

It would be helpful if the authors could compute a thickness image of the specimen expressed in numbers of inelastic mean free paths, i.e., thickness/mean free path = $\ln(I_{\text{total}}/I_0)$, where I_{total} = total intensity in the energy loss spectrum including the zero-loss peak, and I_0 = the zero-loss intensity. This would provide a measure of whether plural inelastic scattering is changing from pixel to pixel in the image. If the thickness map is almost constant, then it might be possible to compute the elemental maps with much less noise, based on the assumption of a constant shape for the background underlying a core edge. In that case, it would be feasible to compute the mean "r-value", i.e., the constant of the inverse power law of the background as a function of energy loss E underlying a particular core edge ($=AE^{-r}$). Then it might be possible to subtract a constant \times the pre-edge background intensity from the post-edge intensity, instead of trying to fit a power law that is unconstrained, and very noisy. This is explained in an early STEM-EELS paper (Leapman R. Scanning Transmission Electron Microscope (STEM) Elemental

Mapping by Electron Energy-Loss Spectroscopy, 1986; Annals of the New York Academy of Sciences 483 (1), 326-338). Although this reference shows that artifacts can be produced from the false assumption that the background shape is constant, there are situations where the assumption could be tested to see if it holds. This reviewer suspects that the present study could be one of those situations. Also, if small numbers of atoms become visible reproducibly, then it's unlikely that the amount of plural scattering could produce a plural inelastic scattering artifact.

We thank the reviewer for raising this point. In our approach, we do not carry out any background subtraction at the level of individual images, as the number of counts would not permit robust statistics; rather, we have carried out background subtraction on a voxel-by-voxel basis on the reconstructed energy-loss volumes. Interpreting the reviewer's comment in this context, we have tested the assumption that the average sample thickness over mean free path, $\ln(I_{\text{total}}/I_0)$, and therefore also the plural scattering contributing to per-voxel spectra, is similar across the different voxels of the reconstructed energy-loss volumes. As far as we understand, this would be the requirement to permit a single value of 'r' for background fitting across all voxels. We have therefore calculated the values of $\ln(I_{\text{total}}/I_0)$ for all voxels. The values vary from 0.26 to 0.31, with the average value for solvent voxels being 0.290 (std 0.006) and for protein voxels being 0.268 (std 0.003). We conclude that there are significant variations of background across different regions of the reconstructed volumes and that we therefore cannot assume a constant background shape.

Other comments:

(1) On page 2, the authors should make it clear that the 3D information about the biomolecular complexes is contained in 2D images of large numbers of identical structures in different orientations in the SPA approach. Some readers who are not familiar with SPA might imagine that a tomographic approach is being taken.

We describe this on page 2 in the paragraph "In SPA, two-dimensional (2D) images with low signal-to-noise ratio (SNR) are combined into a single 3D reconstruction with high SNR. A similar reconstruction technique could conceivably be applied to accumulate spatially resolved elemental signals" and have added the following clarification on page 3: "In this approach, we combine scanning transmission electron microscopy (STEM-) EELS with an SPA-like workflow, in which signal from spectral images of many copies of a macromolecule is accumulated by 3D reconstruction to meet the high dose requirements for elemental mapping."

(2) Typically, the electron dose for SPA analysis is 10-30 electrons/square angstrom. Could the authors comment on the effect of increasing the dose to 100 electrons/square angstrom, for detecting single atoms of metals and other elements in REEL-EM? The resolution achieved in the present study of 20-40 angstroms for the RyR1 and WH structures seems to be quite modest.

We have added the following paragraph to the discussion where we comment on the chosen electron dose:

"For this study we chose a dose of $92 \text{ e}^-/\text{\AA}^2$ to balance the high dose requirements of EELS measurements with the cumulative effects of radiation damage. This exceeds typical doses used in SPA, which are often limited to 30 to $60 \text{ e}^-/\text{\AA}^2$, but is below the doses at which bubble formation causes larger-scale disruption [32]. Future studies could investigate the use of dose weighting [33, 34] to mitigate the effects of radiation damage on resolution although a study in TEM mode suggests that our resolution is not limited by the effects of radiation damage [34]." As Grant and Grigorieff 2015 demonstrate that a resolution of 2.9 Å can be achieved from particles acquired with a dose of $100 \text{ e}^-/\text{\AA}^2$ without dose-filtering, we think that the dose is not the main cause of our modest resolution. Rather, this comes from the fact that we do not have access to phase contrast in our reference images, and likely also from deterioration in the quality of direct alignments over time during automated acquisition.

(3) The authors should include scale bars in all their RyR1 and WH model structures (Fig. 2, Fig. 3, Fig. 5 C,D,E, and Fig. 6).

We have added scale bars for all figures and supplementary figures that show volumes.

(4) This reviewer might have missed seeing it, but the authors should ensure that they include the molecular weights of the RyR1 and WH complexes.

We thank the reviewer for this comment. Both our chosen complexes are large (at around 2.3 and 2.8 MDa, as is now stated in the manuscript). To explain the rationale for this choice in the manuscript, we have added the following paragraph discussing size limitations to the discussion: "Moreover, our reference images are formed by amplitude contrast which is less efficient than phase contrast for SPA samples, which are generally thin and composed of light elements. This limits not only the achievable alignment accuracy but is also likely to pose limitations to the study of small complexes. We circumvented this limitation for this initial study by choosing large complexes (around 2.3 MDa for RyR1 and 2.8 MDa for WH [25]). STEM phase retrieval methods are an active area of investigation [26, 27] and hold potential for improvements in resolution and applicability to small complexes."

(5) It would be helpful if the authors could comment further on the relative advantages of the ADF and EBF images? The authors suggest that neither of these signals provides the contrast seen in standard phase contrast SPA. To what extent can high quality SPA analysis be combined with REEL analysis?

Indeed, neither ADF nor EBF signal are expected to yield a resolution as high as from phase-contrast images. In its current form, we do not anticipate REEL analysis to replace standard SPA analysis of a sample. Rather, we anticipate that both types of analysis could be run in parallel, yielding complimentary information that can be overlaid for accurate atomic modelling. In most cases, the majority of densities seen in high-resolution SPA reconstructions can be accounted for by peptides/nucleic acids with known elemental composition. Additional densities stemming from peptide/nucleic acid components are usually sparse in a reconstruction, although their importance for protein function tends to be unproportionally high. REEL analysis would allow identification of such densities, even if the elemental resolution is lower.

In the manuscript we state this as follows in the discussion section:

"In combination with a high-resolution reconstruction from TEM, even a low-resolution elemental map may be sufficient to assign unknown densities."

We remain open to suggestions if the reviewer/editor feels this should be further clarified.

Reviewer #2:

Remarks to the Author:

Pfeil-Gardiner et al. describe the implementation of electron energy loss spectroscopy with a focused electron beam, scanning over a frozen hydrated cryo-EM specimen in STEM mode. At each pixel position of the electron beam probe, an entire electron energy loss spectrum is recorded with the help of a CEOS energy filter, behind which a Dectris ELA hybrid pixel detector is used to record the spectra. At least initially, a dark-field ring detector was also used to quantify the elastically scattered electrons at higher scattering angles, which give access to a zero-loss HAADF STEM image. However, the authors found that this was very noisy, and a more useful image could be generated by producing a STEM image from the zero-loss electrons in the EELS spectrum. This is then an image, where the absence of electrons indicates the presence of a protein, because at these locations the protein had scattered the electrons out of the EELS spectrum and to higher angles on the HAADF ring detector.

The zero-loss EELS image is first used to reconstruct the structure of the proteins, here using as test specimen the (huge) ryanodine receptor, and also haemoglobin particles. With the help of the particle metadata from that reconstruction, such as particle positions and particle orientations, additional 3D reconstructions for the particles were then generated, using images generated only from electrons that had undergone a specific energy loss. For this, the EELS spectra were sampled in energy bins of 0.77 eV, resulting in sets of images for the spectra that cover -25.4 eV to +770 eV energy loss. These many sets of energy loss images were then used to generate a series of 3D protein reconstructions, each created from the signal within a specific electron energy loss window. Since certain elements in the sample, such as carbon, potassium, nitrogen or oxygen, etc., and also higher weight elements such as phosphor or chloride cause specific energy losses to passing electrons, the images generated from electrons that suffered those characteristic energy losses can then be used to localize those specific atoms in the protein structure. In conclusion, this STEM-EELS method for the first time results in a cryo-EM tool that can localize specific types of atoms in the 3D structure of the single particle proteins.

The obtained resolutions for the two test samples are very modest. The zero-loss EELS images must have a very low signal-to-noise ratio to result in a 24 Å 3D reconstruction from almost half a million particles. One such image is shown in Figure 3B, which illustrates the low SNR of the method. The authors should specify the molecular weight of the imaged particles. The RyR1 is over 3 MDa, if I remember well. What is the weight of the hemoglobin particles? How is the likelihood that this STEM EELS method can also be applied to smaller particles, such as typical 100kD particles that are otherwise investigated a lot in structural biology? This should be discussed in this manuscript.

Ferritin particles filled with an iron core might be an interesting test object for follow-up works (in another paper), or bacteria that accumulate heavy metals, such as uranium. This would then need combination with tomography, but the thicker sample might be well suitable for EELS STEM approaches. Such studies should remain for follow-up works, but could be mentioned here as outlook in the discussion.

A higher resolution might be reachable, if the STEM EELS method would be combined with a phase contrast method, such as defocused CTEM or iDPC imaging. The authors themselves discuss this correctly on Page 8 in lines 10-15. This reviewer, nevertheless, agrees with the authors that this present manuscript is a first proof of concept, which is a milestone achievement by itself and worth its own publication. Combination of the new EELS single particle concept with more routine methods, such as conventional defocused CTEM or the more recent iDPC or ptychography method, is an interesting task for a follow-up projects.

The manuscript is excellently written. It is well structured, interesting to read, has clear figures, and no significant redundancy.

Some details:

Page 2, Introduction: The first paragraph repeats “even though” twice in the same sentence. Please rephrase.

We have rephrased the sentence:

“For protein and nucleotide components, individual atoms can thus be accurately placed, even though experimental data seldom reach atomic resolution and these atoms can be identified, in spite of the fact that no tools exist for elemental mapping.”

Page 2, line 33 (approximately, line numbers were unfortunately missing): You use e^-/A^2 for the first time here. Please define, what that unit refers to. Please also define here, what a normally allowed dose budget would correspond to. You define that later ($100 e^-/A^2$), but it would be more useful to have already here in the text.

We have added a sentence to clarify that $e^-/\text{\AA}^2$ refers to electron dose and to define a normal dose budget:

“Typically, high-resolution studies of biological samples are performed at total electron doses (more accurately, fluences) of 30 to 60 $e^-/\text{\AA}^2$ in order to limit degradation of structural features due to radiation damage.”

Page 3: line 1: You call this method REEL-EM, for “Reconstructed Electron Energy-Loss Elemental Mapping”. I find this is an unfortunate abbreviation, because in the context of this story, EM usually stands for electron microscopy. You define cryo-EM in the first line of the introduction. Now you used REEL-EM here with EM standing for Elemental Mapping. I suggest finding a different way to abbreviate this, or use a different name for the method.

We have changed the name for our method to “REEL analysis” which stands for Reconstructed Electron Energy-Loss Analysis, and have adjusted the manuscript title and all mentions in the text.

Henning Stahlberg.

Reviewer #3:

Remarks to the Author:

A. Summary of key results

The article presents a method for reconstructing 3D spatially resolved EELS data, and thus elemental information, obtained from biological samples.

EELS is an established technique for elemental analysis, however, it requires high doses to achieve sufficient signal and is hence difficult to apply to radiation-sensitive biological samples. The presented technique combines EELS with a single-particle analysis cryoEM workflow, which yields accurate particle coordinates and poses that can be used to extract the energy-loss information from the spectral data. This step distributes the total dose over a large number of particles and thus allows simultaneous spatial and spectral reconstruction from the elastically (zero loss peak) and inelastically scattered electrons, respectively, resulting in a reconstructed 3D volume for each energy bin.

The method is applied to two proteins in a proof-of-principle study for which such 4D datasets including both spatial and spectral information could be obtained. The distribution of abundant elements, such as carbon, could be reconstructed at a resolution sufficient for comparison with higher-resolution structures obtained from conventional cryoEM. The carbon edge could also be discerned in per-voxel spectra, pointing to the possibility of single-atom sensitivity (1-2 C atoms per voxel). Full single-atom sensitivity of non-abundant species could so far not be shown.

B. Originality and Significance

Originality

While previous studies have applied EELS and related techniques to biological specimens, the approach combining EELS and SPA is novel.

Significance

The method addresses an important problem in cryoEM, namely the lack of chemical specificity, which can hinder the accurate assignment of densities especially regarding the localisation of metal ions and small ligands. Hence a technique introducing elemental sensitivity into cryoEM would be highly relevant for many biological applications. While the presented data does not yet address a specific biological question, it serves as a proof-of-principle, thereby demonstrating the potential of the technique. One point to make here is that the elements mapped in this paper

(mainly C, N and O in the solvent), are so abundant in biological specimens, that they will probably not lead to many major insights. This being said, everything has to start somewhere and this is an interesting application.

C. Data and methodology

As this is a proof-of-principle paper, the presented data validates the approach taken. The cryoEM results are validated in the standard ways. The quality of the data set is sufficient to demonstrate that the method works on abundant elements. The ways in which improved data quality could contribute to achieving the goal of single-atom sensitivity for non-abundant elements is discussed.

The presentation of the method and the results is clear.

D. Appropriate use of statistics and treatment of uncertainties

Not applicable?

E. Conclusions: robustness, validity, reliability

The conclusions drawn appear robust, valid and reliable. The current scope of the method is clearly presented and not overstated.

F. Suggested improvements: experiments, data for possible revision

1) To give the reader a better impression of the information contained in the spectral images and how it relates to the information from the bright-field images, the authors should add a larger scale example of a spectral image than the one shown in Fig. 1.

This is a great suggestion. We have added a supplementary figure (Figure S1), in which different energy ranges of an exemplary spectral image are displayed.

2) Supplementary movies: While the changes observed at both the carbon and oxygen edges are very clear, the interpretation of other parts of the spectrum and the corresponding reconstructed volume should be clarified:

- Why do the molecules not appear in the zero-loss peak? Is it because of the DF imaging?

The molecules do appear in the zero-loss peak, however, the contrast is inverted (the zero-loss signal is equivalent to an elastic bright field while energy-loss images show inelastic dark-field signal), so they appear as missing density and are therefore difficult to 'see' in the provided video. For processing of the EBF images, contrast was inverted during processing (Figures 3 and S2), while we did not invert the spectral images for reconstruction of the full spectrum.

- The plasmon peak region is interesting, can the different localisations shown there (also Fig S5) be interpreted in a biologically relevant way?

The plasmon peak can generally show features that are characteristic of certain materials (see Egerton 2011, Chapter 5.2). For biological samples, such features have been described for major components (such as lipid, protein or vitreous ice) (Sun et al 1995). However, there are several challenges in interpreting the plasmon signal. Background subtraction is difficult as material profiles overlap and are not monotonous. Furthermore, the achievable spatial resolution is limited (Egerton 2017). For these reasons we've refrained from a detailed analysis of the plasmon signal.

- In movie 2, both the region between plasmon peak and C edge as well as the trailing C edge itself show interesting geometrical changes, can this be interpreted?

We think that the changes that the reviewer is referring to are likely noise, as they change at every reconstructed energy. The noise appears geometrical due to the applied reconstruction symmetry. In contrast, at elemental edges, the spatial distribution of signal is consistent over several energy bins.

- In Figure 6, the N edge is discernible, although not as strongly as the C edge, in the movie, it is not discernible. Is this due to the remaining carbon signal? If so, can an estimate be made how strong the signal would need to be to be discernible in the movie? A similar question pertains to the P and Cl edges shown in Figure 5.

Yes, this is due to the remaining carbon signal. Although we cannot make an estimate of how much nitrogen would be required for visibility of the nitrogen signal over the carbon signal in the unsubtracted energy-loss reconstruction, we would stress that accurate elemental analysis requires background subtraction. We therefore recommend background subtraction for all elemental maps where pre-edge and edge reconstructions look similar.

Applying background subtraction allowed us to visualize the nitrogen map and its differences from the carbon map (Figure 6 D and E).

3) On page 6, it is stated that “Of the known elements in our sample we detect all except hydrogen, zinc, sulphur, and calcium.” For clarification, it should be stated which elements are present in the sample and/or which elements have been detected.

On page 5, we state which elements are detected as follows:

“Analysing the spectrum summed across all voxels of the RyR1 reconstructions, ionization edges for the most abundant elements are clearly visible: The K-edges of carbon at 284 eV, oxygen at 532 eV (with a small pre-peak indicating the expected presence of gaseous oxygen due to radiation damage [23]), and nitrogen at 401 eV, as well as the L_{2,3}-edges of the buffer components chlorine at 200 eV, potassium at 294 eV, and phosphorus at 132 eV (Figure 5).”

Together with the abovementioned list of undetected elements, we think this clarifies which elements are expected in the sample but are open to reformulating if the reviewer/editor thinks this is helpful.

4) In the same paragraph, parts of Figure 5 seem to have been misreferenced (5B and 5C in the text, whereas the discussion appears to refer to panels 5F and 5G).

We thank the reviewer for pointing this out and have corrected the reference.

5) Discussion of dose at end of the same paragraph (page 6, top): It would be beneficial to add some discussion here (some of it is addressed in the Discussion section) regarding how these inaccuracies in the poses could be improved, e.g. the role of higher accumulated dose (larger datasets), would lower dose per particle help with pose accuracy? See also 9).

We have extended the discussion section to discuss the chosen dose per particle:

“For this study we chose a dose of 92 e⁻/Å² to balance the high dose requirements of EELS measurements with the cumulative effects of radiation damage. This exceeds typical doses used in SPA, which are often limited to 30 to 60 e⁻/Å², but is below the doses at which bubble formation causes larger-scale disruption [32]. Future studies could investigate the use of dose weighting [33, 34] to mitigate the effects of radiation damage on resolution although a study in TEM mode suggests that our resolution is not limited by the effects of radiation damage [34].”

6) Single-voxel spectra (Fig 5, Fig 6): The presented voxels in Fig 5 and 6 appear to be the same. If so, this should be stated. How many voxels have been studied in this way? Are they all

featuring similar characteristics? Could conclusions be drawn from comparisons, e.g. between a voxel in the protein bulk and the micelle region, about abundance of nitrogen, and hence, by extrapolation, about the number of atoms of a species in a given set of neighbouring voxels to get a reliable edge signal?

The single-voxel spectra in Figures 5 and 6 are not from the same positions. They were chosen as random examples for voxels corresponding to positions in the protein. We see similar results throughout the protein (we studied at least 20 blindly chosen voxel positions). For voxels outside of the protein, the carbon edge is less visible (again, we studied at least 20 blindly chosen voxel positions).

From a single voxel, it is not possible to draw conclusions about nitrogen abundance as the corresponding edge is not visible above noise. Extending this analysis further, we tested whether quantitative analysis is feasible for mean spectra of several voxels, by calculating the mean spectra for all voxels in the micelle and all voxels in the protein. Unfortunately, as only few voxels contribute to the micelle, the SNR in the mean micelle spectrum is not sufficient to reliably measure the height of the nitrogen edge (small differences in the measurement of the edge height lead to large variations in the ratios). A quantitative analysis is therefore not feasible with the current data, although qualitatively, we see that the ratio of the carbon to nitrogen edge heights is larger for the micelle than for the protein, as we would expect.

7) In Figure S6, the grey area on the spectrum extends further than the range shown in the inset.

We thank the reviewer for pointing this out and have corrected the figure.

8) The Discussion section is largely a summary plus an outlook regarding some of the current problems and the future development of the method, given that, the authors may want to consider renaming this section. As already suggested in point 5), the authors may further want to consider moving the discussion regarding total dose and pose inaccuracies to the relevant place in Section Reconstruction of energy-loss data generates 3D EELS maps (page 6) and focus on future development in the section currently named Discussion.

We thank the reviewer for bringing this to our attention. We have extended the discussion about single-atom sensitivity and moved it from the results section to the discussion section. It now reads:

“With 460,651 4-fold symmetric particles, collected at a dose of $92 \text{ e}^-/\text{\AA}^2$, the effective accumulated dose for the RyR1 reconstruction is around $1.7 \cdot 10^8 \text{ e}^-/\text{\AA}^2$ which exceeds a previously estimated minimum dose of $10^7 \text{ e}^-/\text{\AA}^2$ for single-atom sensitivity [3] by more than an order of magnitude. It is important to note that this estimate was performed on the basis of dry protein on carbon foil (thickness 4 nm) for which less background is expected than for our ice-embedded samples (thickness > 30 nm), correspondingly increasing the required total dose to achieve similar SNR. Moreover, the estimate referred to elemental detection in a single exposure. For single-particle averaging techniques, imperfect alignment accuracy increases the required cumulative dose. To distinguish between the effects of cumulative dose and the effects of alignment accuracy on sensitivity, an important observation is that, for a single voxel in the protein, containing 1-2 carbon atoms, the corresponding spectrum features a discernible carbon K-edge. The carbon signal is not strongly affected by alignment, since neighbouring voxels in the protein also contain carbon; therefore, the detection of carbon but not less abundant elements in single-voxel spectra points to an important role of alignment inaccuracies in limiting the current data set.”

Together with several further additions to the discussion section in the resubmitted manuscript, as described in various points above, we think that this is now appropriately titled.

9) Regarding the future development of the technique, the authors list many different approaches. It would be beneficial to clarify which of these the authors think are the most likely to significantly improve the method in the near future.

We have expanded upon a sentence in the concluding paragraph: “They indicate that increased image quality and dataset size are the most important avenues for further improvements in resolution and sensitivity.”

10) Choice of test samples: were there particular reasons to choose these proteins as test samples (other than the presence of metal ligands)? Would it make sense to test the method on standard cryoEM samples? To test sensitivity, could proteins with more than one metal ligand at a binding site be used?

We consider worm hemoglobin to be a standard test sample. It has been used in previous methodological publications (for example Ravelli et al 2020, Afanasyev et al 2017, Schatz et al 1992). More commonly, apoferritin is used as a standard test sample in cryo-EM. It performs well at high refinement resolution due to high symmetry and rigidity, making it ideal for many studies. In our method, we are limited to lower resolutions than most SPA studies. At low resolutions, apoferritin appears as a sphere, making it very difficult to refine reconstruction poses. We therefore chose samples that show distinct geometries for different projection views also at low resolutions.

Indeed, proteins with more than one metal at a binding site could help investigating sensitivity in future studies.

11) References 34 and 35 appear to be the same.

One of the references refers to a PDB entry, the other to the corresponding publication. We are following the citation recommendations from <https://www.wwpdb.org/about/cite-us> by citing both the structure with its DOI and the source publication.

G. References: appropriate credit to previous work?

Largely, the reference list gives a good picture of past work. However, the authors could perhaps cite Herdman et al Structure 2022 where single-particle EM was used to map single Holmium/Calcium ions. Another article to perhaps cite is Wu et al Science 2012, where beam damage was used to distinguish between protein and DNA.

We have added the reference to Herdman et al Structure 2022 to the introduction section (reference 17).

H. Clarity and context: lucidity of abstract/summary, appropriateness of abstract, introduction and conclusions

Abstract, Introduction and conclusions are all appropriate and clear. As the section titled Discussion is mostly contains a summary, conclusions and an outlook, the authors may want to consider renaming this section as suggested above.

See 8) above.

Reviewer #4:
None

Editor's note: Reviewer #4 co-reviewed with Reviewer #3 and therefore did not provide separate comments to the authors.

References

- Afanasyev, P. et al "Single-particle cryo-EM using alignment by classification (ABC): the structure of *Lumbricus terrestris* haemoglobin", IUCrJ, 2017
- Egerton, R. F., "Electron Energy-Loss Spectroscopy in the Electron Microscope, Third edition, 2011 (book)
- Egerton, R. F., "Scattering delocalization and radiation damage in STEM-EELS", Ultramicroscopy, 2017
- Grant, T. and Grigorieff, N., "Measuring the optimal exposure for single particle cryo-EM using a 2.6 Å reconstruction of rotavirus VP6", 2015
- Ravelli, R. et al, "Cryo-EM structures from sub-nl volumes using pin-printing and jet vitrification", Nature Communications, 2020
- Schatz, M. and Heel, M., "Invariant recognition of molecular projections in vitreous ice preparations", Ultramicroscopy 1992
- Sun, S. Q. et al, "Quantitative water mapping of cryosectioned cells by electron energy-loss spectroscopy", Journal of Microscopy, 1995
- Vilas, J. L. et al, "Emerging Themes in CryoEM–Single Particle Analysis Image Processing", Chemical Reviews, 2022

Response to Reviewers

We thank the reviewers for their positive evaluation of our revised manuscript and have addressed the remaining minor point in the final submitted version, as described below.

Reviewer #1:

Remarks to the Author:

The authors have satisfactorily addressed the questions raised by this reviewer.

Although the current paper describes a proof-of-principle technique, it could become a valuable approach and so is recommended for publication.

Thank you!

Reviewer #2:

Remarks to the Author:

The authors have addressed all my earlier comments to my fullest satisfaction.

Thank you!

Reviewer #3:

Remarks to the Author:

I recommend publication of the manuscript.

Thank you!

Reviewer #4:

Remarks to the Author:

The authors have addressed all points raised by the reviewers well and have made appropriate changes. The present version of the manuscript is suitable for publication.

Thank you!

As a very minor point, the authors may want to consider mentioning the statistics on the consistency of the single-voxel spectra across the protein reported in their response to Reviewer #3, point 6 either in the main text, figure caption or methods section as this would strengthen the point they are making.

Thank you for pointing this out, we have included these statistics in the 'Statistics and Reproducibility' section in the methods.